# CAN WATERMARKED LLMS BE IDENTIFIED BY USERS VIA CRAFTED PROMPTS?

**Aiwei Liu[1], Sheng Guan[2], Yiming Liu[1], Leyi Pan[1], Yifei Zhang[3], Liancheng Fang[4], Lijie Wen[1] *, Philip S. Yu[4], Xuming Hu[5]**

[1] Tsinghua University    [2] Beijing University of Posts and Telecommunications
[3] The Chinese University of Hong Kong    [4] University of Illinois at Chicago
[5] Hongkong University of Science and Technology (Guangzhou)

liuaw20@mails.tsinghua.edu.cn, guansheng2022@bupt.edu.cn, wenlj@tsinghua.edu.cn

[Official]: https://github.com/THU-BPM/Watermarked_LLM_Identification

## ABSTRACT

Text watermarking for Large Language Models (LLMs) has made significant progress in detecting LLM outputs and preventing misuse. Current watermarking techniques offer high detectability, minimal impact on text quality, and robustness to text editing. However, current researches lack investigation into the imperceptibility of watermarking techniques in LLM services. This is crucial as LLM providers may not want to disclose the presence of watermarks in real-world scenarios, as it could reduce user willingness to use the service and make watermarks more vulnerable to attacks. This work investigates the imperceptibility of watermarked LLMs. We design the first unified identification method called `Water-Probe` that identifies all kinds of watermarking in LLMs through well-designed prompts. Our key motivation is that current watermarked LLMs expose consistent biases under the same watermark key, resulting in similar differences across prompts under different watermark keys. Experiments show that almost all mainstream watermarking algorithms are easily identified with our well-designed prompts, while `Water-Probe` demonstrates a minimal false positive rate for non-watermarked LLMs. Finally, we propose that the key to enhancing the imperceptibility of watermarked LLMs is to increase the randomness of watermark key selection. Based on this, we introduce the `Water-Bag` strategy, which significantly improves watermark imperceptibility by merging multiple watermark keys.

## 1 INTRODUCTION

The rapid advancement of large language models (LLMs) has led to remarkable achievements in tasks such as question answering (Zhuang et al., 2024), programming (Jiang et al., 2024b), and reasoning (Wei et al., 2022), with widespread applications across various scenarios. However, the extensive use of LLMs has also raised concerns regarding copyright protection and misuse. Recent research indicates that malicious attackers can steal LLMs through model extraction techniques (Yao et al., 2024), and some users may abuse LLMs to generate and spread harmful information (Wei et al., 2024).

Text watermarking techniques for LLMs have become an important method to mitigate the above issues by adding detectable features to LLM outputs (Liu et al., 2024b). Recent researches on LLM watermarking have focused on improving watermark detectability (Kirchenbauer et al., 2023a), minimizing impact on generated text (Aaronson & Kirchner, 2022), and enhancing robustness against text modifications (Liu et al., 2024a). However, no work has considered the imperceptibility of watermarked LLMs, i.e., whether users can know if an LLM service is watermarked. In real-world scenarios, LLM service providers may not disclose the existence of watermarks, as it could reduce user willingness to use the service and make it more vulnerable to attacks (Sadasivan et al., 2023). As more LLM services consider implementing watermarks, it is crucial to investigate whether users can identify watermarked LLMs solely through crafted prompts.

---

*Corresponding author

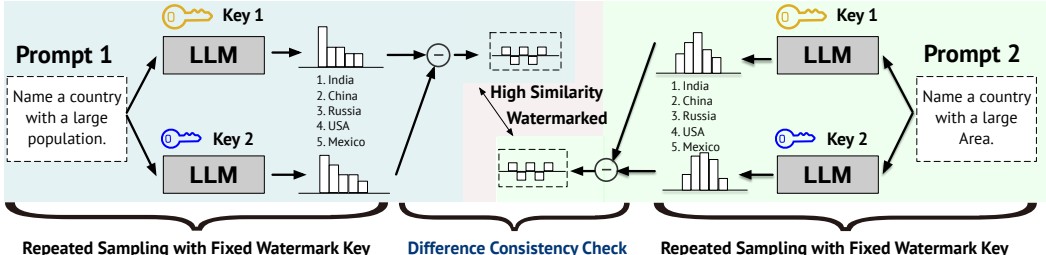

Figure 1: Illustration of our `Water-Probe` algorithm for identifying watermarked LLMs. We first construct two prompts with similar output distributions, then sample repeatedly using two fixed watermark keys for each prompt. The presence of a watermark is determined by comparing the similarity of distribution differences between the two prompts. Details in §3.

Some studies focus on the imperceptibility of watermarked text, ensuring watermarked and non-watermarked texts are indistinguishable (Hu et al., 2023; Wu et al., 2023b). However, even if individual watermarked texts are imperceptible, the distribution of numerous watermarked texts may reveal whether the LLM is watermarked, especially when repeatedly sampling with the same watermark key (Wu et al., 2024). While some studies explore cracking watermarks using large volumes of watermarked text (Jovanović et al., 2024; Sadasivan et al., 2023; Wu & Chandrasekaran, 2024), they assume the LLM is watermarked and cannot determine if the LLM is watermarked. The most relevant work is Gloaguen et al. (2024), which proposes a black-box detection method for watermarked LLMs. However, their approach uses different detection methods for different watermarks and cannot effectively detect all watermarking techniques.

In this work, we propose `Water-Probe`, the first unified method for identifying watermarked LLMs that can detect all types of watermarks embedded during the LLM's text generation process. (see related work section for this type of watermarking) Our motivation stems from a key observation: all current LLM watermarking algorithms expose consistent bias when repeatedly sampled under the same watermark key. Based on this, our `Water-Probe` algorithm first crafts prompts to perform repeated sampling under the same watermark key, then compares the consistency of sampling distribution differences across different prompts under a pair of watermark keys. Highly consistent differences indicate a watermarked LLM.

In our experiments, we demonstrate that the `Water-Probe` algorithm achieves high accuracy in detecting various types of watermarked LLMs. We also show its applicability across different LLMs, maintaining a low false positive rate for non-watermarked LLMs. Furthermore, our algorithm exhibits robust performance across different sampling methods and temperature settings.

Finally, we explore methods to enhance the imperceptibility of watermarked LLMs. We find that increasing the randomness of watermark key selection is crucial, as it makes it more difficult to construct prompts for repeated sampling using the same key. Based on this, we propose the `Water-Bag` algorithm, which combines multiple watermark keys into one, randomly selecting a key for each generation and choosing the highest score for detection. Although increasing key selection randomness often leads to a slight decrease in detectability, it significantly enhances the imperceptibility of watermarked LLMs. Addressing this trade-off should be an important direction for future work.

Our main contributions are summarized as follows:

- We propose `Water-Probe`, the first unified algorithm that can detect various types of watermarked LLMs by analyzing the consistency of sampling distribution differences across different prompts under fixed watermark keys.

- Through extensive experiments, we demonstrate that `Water-Probe` achieves high detection accuracy across different LLMs, watermarking methods, and sampling settings, while maintaining a low false positive rate for non-watermarked LLMs.

- We introduce Water-Bag, a novel algorithm that enhances LLM watermark imperceptibility by combining multiple watermark keys, and analyze the trade-off between watermark detectability and imperceptibility.

## 2 PRELIMINARIES

**Definition 1** (Large Language Model). *An LLM $M$ is a function that, given an input $x$ and a partial output sequence $y_{1:i-1}$, produces a probability distribution $P_M(y_i|x, y_{1:i-1})$ over possible next tokens $y_i$. The model generates complete outputs by iteratively sampling from these distributions.*

Since this work focuses solely on watermarks embedded in LLM services, we will only consider the **watermark during generation** type mentioned in §6. The definition of the watermark rule is given below.

**Definition 2** (Watermark Rule). *A watermark rule is typically a function $F$ that adjusts the current LLM's predicted probability distribution based on a watermark key $k$ to obtain a new probability distribution. Formally, given an LLM $P_M$ and a key $k$, the watermark rule $F$ modifies the distribution as follows:*

$$P_M^F(y_i|x, y_{1:i-1}, k) = F(P_M(y_i|x, y_{1:i-1}), k) \tag{1}$$

*where $P_M^F$ is the modified probability distribution for the next token $y_i$.*

The main difference between watermarking algorithms lies in how they determine the watermark key. Based on this, we categorize watermarking algorithms into **n-gram based watermarking** and **fixed-key-list based watermarking**. We will now introduce these two types of watermarking algorithms.

**Definition 3** (N-Gram Based Watermarking). *In n-gram based watermarking, the watermark key $k_i$ for generating the current token $y_i$ is determined by a function $f$ that takes two inputs:*

$$k_i = f(K, y_{i-n:i-1}) \tag{2}$$

*where $K$ is a pre-selected master key, and $y_{i-n:i-1}$ represents the previous $n$ tokens.*

N-gram based watermarking ensures that for the same n-token prefix, the watermark key for generating the next token remains consistent. This approach is widely used in current watermarking algorithms, including KGW (Kirchenbauer et al., 2023a), KGW-V2 (Kirchenbauer et al., 2023b), Aar (Aaronson & Kirchner, 2022), DiPmark (Wu et al., 2023b), and SIR (Liu et al., 2024a). Next, we define fixed-key-list-based watermarking:

**Definition 4** (Fixed-Key-List Based Watermarking). *Let $K = \{k_1, k_2, ..., k_m\}$ be a fixed key list. For a given starting index $s \in \{1, ..., m\}$, the watermark key $k_i$ for generating the $i$-th token is:*

$$k_i = k_{((s+i-1) \bmod m)+1} \tag{3}$$

*where the starting index $s$ may be randomly chosen for each generation process.*

This approach is employed in algorithms such as Exp-Edit (Kuditipudi et al., 2023), where keys are used sequentially from a potentially random starting position in the key list.

To formalize our approach, we define our goal as follows:

**Definition 5** (Black-box Watermark Identification). *A function $\mathcal{D} : P_M \rightarrow \{0, 1\}$ that classifies a language model $P_M$ as watermarked (1) or not (0), without access to its internal parameters.*

## 3 WATERMARKED LLM IDENTIFICATION

### 3.1 WHY WATERMARKED LLM IDENTIFICATION IS POSSIBLE

Definition 2 implies that LLMs typically introduce some distortion to the distribution. However, there exist distortion-free watermarking algorithms that satisfy Definition 2. We first provide a definition and then demonstrate that LLMs with distortion-free watermarks can still be identified.

**Definition 6** (Distortion-Free Watermark). *A watermarking algorithm is considered distortion-free if, for all possible inputs $x$ and partial output sequences $y_{1:i-1}$, the expected output distribution of the watermarked model $P_M^F$ over all possible watermark keys $k \in \mathcal{K}$ is identical to the original model $P_M$:*

$$\mathbb{E}_{k \in \mathcal{K}}[P_M^F(y_i|x, y_{1:i-1}, k)] = P_M(y_i|x, y_{1:i-1}). \tag{4}$$

This equation indicates that the expected output distribution of watermarked text remains unchanged when the watermark key is randomly selected across all possible keys. However, as shown in Equation 1, sampling with a specific watermark key $k$ introduces a difference between $P_M^F(y_i|x, y_{1:i-1}, k)$ and $P_M(y_i|x, y_{1:i-1})$. This observation leads to the following theorem on detectability of watermarked LLMs:

**Observation 1** (Distributional Difference of Watermarked LLMs). *Let $P_M$ be a language model and $F$ a watermark rule as defined in Definition 2. For a given watermark key $k$, the probability distribution of the watermarked model $P_M^F(y_i|x, y_{1:i-1}, k)$ differs from the original distribution $P_M(y_i|x, y_{1:i-1})$. This distributional difference suggests the potential detectability of the watermark.*

### 3.2 Pipeline of Watermarked LLM Identification

Observation 1 implies that the key to identifying a watermarked LLM is to construct prompts that allow for multiple samplings using the same watermark key to reveal the difference between $P_M^F(y_i|x, y_{1:i-1}, k)$ and $P_M(y_i|x, y_{1:i-1})$. However, due to the black-box setting, we cannot directly access the origin logits $P_M(y_i|x, y_{1:i-1})$. Instead, we calculate the difference in LLM outputs for two distinct keys, defined as $\Delta(x, k_m, k_n) = P_M^F(\cdot|x, k_m) - P_M^F(\cdot|x, k_n)$, where we use $P_M^F(\cdot|x, k)$ to represent the output distribution.

If two prompts yield similar output distributions, the same watermark key should have similar effects on both prompts (proven in Theorem 1). We determine if an LLM contains a watermark by comparing the consistency of the effects of two watermark keys on two similar prompts. Specifically, given $x_1$, $x_2$, $k_1$, and $k_2$, we assess the similarity between $\Delta(x_1, k_1, k_2)$ and $\Delta(x_2, k_1, k_2)$. High similarity indicates the presence of a watermark; otherwise, we conclude there is no watermark. Based on the above analysis, we now present the process of the `Water-Probe` algorithm.

**Step 1: Construct highly correlated prompts.** Construct $N$ prompts $x_1, x_2, ..., x_N$ such that their output probability distributions under $M$ are highly similar, which can be expressed as:

$$\forall i, j \in \{1, 2, ..., N\}, \text{KL}(P_M(\cdot|x_i)||P_M(\cdot|x_j)) \leq \epsilon \text{ and } x_i \neq x_j \tag{5}$$

where $\text{KL}(\cdot||\cdot)$ is the Kullback-Leibler divergence, $P_M(\cdot|x_i)$ is the output probability distribution for prompt $x_i$ under the LLM $M$, and $\epsilon$ is a small threshold indicating high similarity between distributions.

**Step 2: Sampling with simulated fixed watermark keys.** Since we cannot access the logits under a given watermark key, we need to use repeated sampling to estimate the distribution. We construct a set of simulated watermark keys $K = \{k_1, k_2, ..., k_m\}$ based on our prompt design (detailed in subsequent sections). For each prompt $x_i$ and each simulated key $k_j \in K$, we estimate the distribution as follows:

$$\hat{P}_M^F(y|x_i, k_j) = \frac{1}{W} \sum_{w=1}^{W} \mathbf{1}_{y_{i,j}^w = y}, \quad \text{where } y_{i,j}^w \sim P_M^F(y|x_i, k_j) \tag{6}$$

where $W$ is the sample count, $\mathbf{1}_A$ is the indicator function, and $y_{i,j}^w$ is the $w$-th sample sampled from $P_M^F(y|x_i, k_j)$. Specific prompt techniques for different watermarking algorithms will be detailed later. Note that our prompt design for simulating watermark keys assumes the target LLM has a watermark. If it doesn't, then $P_M^F(y|x_i, k_j) = P_M(y|x_i)$ for all simulated keys.

**Step 3: Analyze Cross-Prompt Watermark Consistency.** We first assume that the watermark rule satisfies Lipschitz continuity. Based on this assumption, we can deduce that the differences in output distributions produced by a watermark key pair for highly correlated prompts are similar.

**Assumption 1** (Lipschitz Continuity of Watermark Rule). *For prompts $x_1$ and $x_2$ satisfying the similarity condition in Equation 5, the watermark rule $F$ satisfies Lipschitz continuity. That is, there*

*exists a constant $L > 0$ such that for the probability distributions $P_M(\cdot|x_1), P_M(\cdot|x_2)$ and any watermark key $k \in \mathcal{K}$:*

$$\|F(P_M(\cdot|x_1), k) - F(P_M(\cdot|x_2), k)\|_1 \leq L \cdot \|P_M(\cdot|x_1) - P_M(\cdot|x_2)\|_1. \tag{7}$$

**Theorem 1** (Consistency of Watermark Effect). *Let $x_1$ and $x_2$ be two different prompts satisfying the similarity condition in Equation 5. Let $k_1$ and $k_2$ be two randomly sampled watermark keys from the key space $\mathcal{K}$. The effect of applying these keys on the output distribution should be highly consistent across prompts:*

$$\mathbb{E}_{k_1, k_2 \sim \mathcal{K}}[Sim(P_M^F(\cdot|x_1, k_1) - P_M^F(\cdot|x_1, k_2), P_M^F(\cdot|x_2, k_1) - P_M^F(\cdot|x_2, k_2))] \geq \rho \tag{8}$$

*where $P_M^F$ is the watermarked distribution, and $Sim(\cdot, \cdot)$ is a similarity measure (e.g., cosine similarity), and $\rho$ should be a constant significantly greater than 0.*

Theorem 1 implies that for prompts with similar output distributions under the LLM, the expected differences introduced by the two watermark keys should also be similar. The formal proof is provided in Appendix A.

Based on Theorem 1, we calculate the average similarity using the estimated distributions from Step 2. To ensure stability across different sampling temperatures, we first apply a rank transformation. Specifically, for a token $y$, its rank is defined as the number of tokens with probability greater than or equal to that of $y$, denoted as $R(P(y|x)) = |\{y' \in V : P(y'|x) \geq P(y|x)\}|$. We then compute the expected similarity:

$$\bar{S} = \frac{1}{N} \sum_{x_i \neq x_j \in \mathcal{X}} \sum_{k_m \neq k_n \in \mathcal{K}} \text{Sim}(\Delta_R(x_i, k_m, k_n), \Delta_R(x_j, k_m, k_n)) \tag{9}$$

Here, $\mathcal{X}$ is the prompt set, $\mathcal{K}$ is the watermark key set, $N = |\mathcal{X}|(|\mathcal{X}| - 1)|\mathcal{K}|(|\mathcal{K}| - 1)$, and $\Delta_R(x, k_m, k_n) = R(\hat{P}_M^F(\cdot|x, k_m)) - R(\hat{P}_M^F(\cdot|x, k_n))$. We verify the importance of rank transformation in Appendix G.

According to Theorem 1, if $M$ contains a watermark, the similarity obtained from Equation 9 should be significantly greater than 0. If $M$ does not contain a watermark, we assume $P_M^F = P_M$ for any $k$, so Equation 9 should represent the similarity between two random vectors with zero mean, which should be close to 0. A detailed analysis of the no-watermark case is provided in Appendix B.

Based on this, we design the following z-test to perform hypothesis testing on the average similarity:

$$z = (\bar{S} - \mu)/\sigma \tag{10}$$

where $\bar{S}$ is the observed average similarity, $\sigma$ is the standard deviation of the $\bar{S}$, and $\mu$ is the mean of the $\bar{S}$ under the no-watermark case. Theoretically, $\mu$ should be 0 for an unwatermarked LLM. However, in practice, we may choose a value slightly greater than 0 to account for potential biases introduced by our prompt construction method or other factors. The standard deviation $\sigma$ is estimated through multiple experiments.

We reject the null hypothesis (no watermark) and conclude the LLM is likely watermarked if: $z > z_\alpha$, where $z_\alpha$ is the chosen significance level $\alpha$. In this work, we consider a z-score between 4 and 10 as moderate confidence, and above 10 as high confidence.

### 3.3 CONSTRUCTING REPEATED SAMPLING WITH SAME WATERMARK KEY

In the previous section, we introduced the basic pipeline of our `Water-Probe` algorithm. A key challenge is constructing prompts that enable multiple samplings with the same watermark key. This varies for different watermarking algorithms. We'll discuss approaches for n-gram based and fixed-key-list based methods.

For **N-gram based watermarking** (Definition 3), since the watermark key is derived from the previous N tokens, we can design prompts that make the LLM generate N irrelevant tokens before following the prompt. An example is provided below:

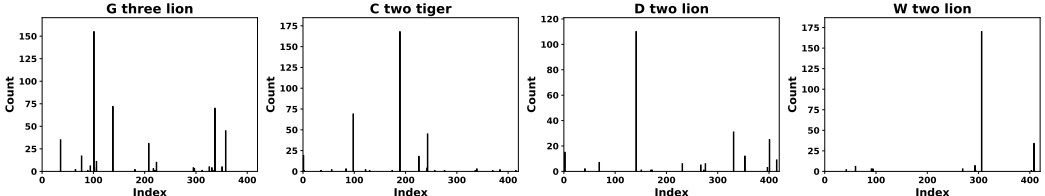

Figure 2: Distribution of start keys for identical prefixes in **Exp-Edit** watermarking. Analysis based on prompts described in Section 3.3 for `Watermark-Probe-v2`. Each subplot represents a specific prefix(in title).

> **Prompt 1: Example Prompt for `Watermark-Probe-v1`**
>
> Please generate *abcd* before answering the question.
> **Question:** Name a country with a large population.
> **Answer:** *abcd* India

In the example above, we assume generating *abcd* does not affect the distribution for answering the question. However, in practice, it's challenging to ensure completely irrelevant tokens. Consequently, the $\bar{S}$ for an unwatermarked LLM constructed this way may be slightly above 0. We refer to the `Watermark-Probe` using prompts similar to the above table as `Watermark-Probe-v1`.

For **fixed-key-set based watermarking**, since the start watermark key is randomly selected each time, our approach is to approximate multiple samplings with the same watermark key by exploiting the correlation between the watermark key and the generated tokens. Specifically, we prompt the LLM to perform some quasi-random generation initially. Generally, the same watermark start key will only generate a few fixed sampling results, so we can assume that identical sampling result prefixes are generated by the same watermark key. Here's a specific example of how we construct prompts for this approach:

> **Prompt 2: Example Prompt for `Watermark-Probe-v2`**
>
> Please generate a sentence that satisfies the following conditions: The first word is randomly sampled from *A-Z*. The second word is randomly sampled from *zero to nine*. The third word is randomly sampled from *cat, dog, tiger and lion*. Then answer the question: Name a country with a large population.
> **Answer:** *A one cat* China

As shown in the example above, we first prompt the LLM to generate N (3 in this case) random tokens before answering the question. Different random token combinations typically correspond to a few watermark keys. Figure 2 illustrates the distribution of watermark key counts for varying numbers of random tokens. As evident from the figure, given a fixed prefix, the vast majority of cases utilize a specific key. So this approach thus approximates sampling with the same watermark key. Similarly, we refer to the prompting method in the above table as `Watermark-Probe-v2`.

We provide the detailed steps of the Water-Probe algorithm in Algorithm 1 in the appendix.

## 4 EXPERIMENT ON WATERMARKED LLM IDENTIFICATION

### 4.1 EXPERIMENT SETUP

**Tested Watermarking Algorithms**: We evaluated a diverse range of LLM watermarking algorithms, including N-Gram based watermarking and Fixed-Key-List based watermarking. For N-Gram based watermarking, we tested KGW (Kirchenbauer et al., 2023a) ($\gamma = 0.5$, $\delta = 2$), Aar (Aaronson & Kirchner, 2022) ($N = 1$), KGW-Min Kirchenbauer et al. (2023b) (window size of 4), KGW-Skip (Kirchenbauer et al., 2023b) (window of 3), DiPMark (Wu et al., 2023b) ($\alpha = 0.45$), and $\gamma$ reweighting (Hu et al., 2023). For Fixed-Key-List based watermarking, we examined EXP-Edit (Kuditipudi et al., 2023) and ITS-Edit (Kuditipudi et al., 2023), both with a key length of 420. Details of these algorithms are provided in Appendix F.

Table 1: Detection similarities for various LLMs with and without different watermarks, calculated using Equation 9 and our two identification methods: `Water-Probe-v1` and `Water-Probe-v2`. ▮ indicates high-confidence watermark identification and ▮ indicates low-confidence watermark identification while no color indicates no watermark identified. The corresponding z-scores can be found in Table 7 in the Appendix.

| LLM | N-Gram | | | | | | | Fixed-Key-List | |
|---|---|---|---|---|---|---|---|---|---|
| | Non | KGW | Aar | KGW-Min | KGW-Skip | DiPmark | $\gamma$-Reweight | EXP-Edit | ITS-Edit |
| **`Water-Probe-v1` (w. prompt 1)** | | | | | | | | | |
| *Qwen2.5-1.5B* | 0.02 ±0.02 | 0.37 ±0.02 | 0.88 ±0.06 | 0.37 ±0.02 | 0.39 ±0.01 | 0.55 ±0.01 | 0.55 ±0.01 | 0.01 ±0.02 | 0.00 ±0.04 |
| *OPT-2.7B* | 0.05 ±0.01 | 0.47 ±0.01 | 0.91 ±0.01 | 0.42 ±0.02 | 0.45 ±0.01 | 0.60 ±0.01 | 0.61 ±0.01 | 0.08 ±0.02 | 0.09 ±0.01 |
| *Llama-3.2-3B* | 0.04 ±0.02 | 0.53 ±0.01 | 0.90 ±0.01 | 0.48 ±0.00 | 0.49 ±0.01 | 0.61 ±0.01 | 0.61 ±0.01 | 0.03 ±0.01 | 0.04 ±0.01 |
| *Qwen2.5-3B* | 0.03 ±0.01 | 0.33 ±0.02 | 0.75 ±0.05 | 0.33 ±0.02 | 0.38 ±0.00 | 0.51 ±0.01 | 0.53 ±0.01 | 0.03 ±0.01 | 0.06 ±0.02 |
| *Llama2-7B* | 0.02 ±0.01 | 0.42 ±0.01 | 0.87 ±0.01 | 0.31 ±0.01 | 0.42 ±0.01 | 0.56 ±0.01 | 0.56 ±0.04 | 0.03 ±0.02 | 0.02 ±0.00 |
| *Mixtral-7B* | 0.01 ±0.02 | 0.41 ±0.01 | 0.85 ±0.02 | 0.37 ±0.01 | 0.41 ±0.02 | 0.57 ±0.01 | 0.58 ±0.03 | 0.00 ±0.00 | 0.02 ±0.02 |
| *Qwen2.5-7B* | 0.07 ±0.04 | 0.41 ±0.02 | 0.82 ±0.02 | 0.34 ±0.03 | 0.38 ±0.02 | 0.43 ±0.03 | 0.43 ±0.02 | 0.06 ±0.01 | 0.04 ±0.02 |
| *Llama-3.1-8B* | 0.01 ±0.02 | 0.41 ±0.02 | 0.85 ±0.02 | 0.41 ±0.01 | 0.39 ±0.01 | 0.57 ±0.02 | 0.58 ±0.00 | 0.02 ±0.02 | 0.00 ±0.01 |
| *Llama2-13B* | 0.01 ±0.03 | 0.41 ±0.01 | 0.86 ±0.01 | 0.31 ±0.02 | 0.40 ±0.02 | 0.58 ±0.02 | 0.60 ±0.01 | 0.02 ±0.01 | 0.02 ±0.03 |
| ***Average*** | 0.029 | 0.418 | **0.854** | 0.371 | 0.412 | 0.553 | 0.505 | 0.031 | 0.032 |
| **`Water-Probe-v2` (w. prompt 2)** | | | | | | | | | |
| *Qwen2.5-1.5B* | 0.02 ±0.02 | 0.30 ±0.01 | 0.83 ±0.01 | 0.29 ±0.01 | 0.27 ±0.02 | 0.49 ±0.02 | 0.52 ±0.03 | 0.39 ±0.03 | 0.60 ±0.00 |
| *OPT-2.7B* | 0.04 ±0.03 | 0.29 ±0.02 | 0.88 ±0.01 | 0.23 ±0.01 | 0.19 ±0.02 | 0.42 ±0.02 | 0.43 ±0.03 | 0.43 ±0.01 | 0.62 ±0.00 |
| *Llama-3.2-3B* | 0.00 ±0.01 | 0.31 ±0.01 | 0.89 ±0.01 | 0.33 ±0.00 | 0.24 ±0.01 | 0.51 ±0.01 | 0.54 ±0.01 | 0.52 ±0.01 | 0.84 ±0.00 |
| *Qwen2.5-3B* | 0.03 ±0.02 | 0.35 ±0.04 | 0.78 ±0.01 | 0.29 ±0.02 | 0.28 ±0.01 | 0.45 ±0.02 | 0.45 ±0.02 | 0.39 ±0.02 | 0.71 ±0.00 |
| *Llama2-7B* | 0.04 ±0.02 | 0.34 ±0.01 | 0.82 ±0.02 | 0.33 ±0.01 | 0.28 ±0.01 | 0.50 ±0.01 | 0.51 ±0.02 | 0.48 ±0.01 | 0.81 ±0.00 |
| *Mixtral-7B* | 0.09 ±0.01 | 0.34 ±0.04 | 0.83 ±0.01 | 0.29 ±0.02 | 0.24 ±0.01 | 0.51 ±0.01 | 0.53 ±0.00 | 0.42 ±0.02 | 0.81 ±0.00 |
| *Qwen2.5-7B* | -0.01 ±0.04 | 0.26 ±0.02 | 0.70 ±0.00 | 0.28 ±0.02 | 0.23 ±0.01 | 0.32 ±0.03 | 0.35 ±0.02 | 0.32 ±0.02 | 0.73 ±0.00 |
| *Llama-3.1-8B* | 0.01 ±0.00 | 0.31 ±0.01 | 0.77 ±0.01 | 0.29 ±0.02 | 0.26 ±0.00 | 0.50 ±0.01 | 0.51 ±0.01 | 0.43 ±0.01 | 0.71 ±0.00 |
| *Llama2-13B* | 0.01 ±0.02 | 0.35 ±0.01 | 0.82 ±0.02 | 0.26 ±0.02 | 0.26 ±0.01 | 0.50 ±0.01 | 0.53 ±0.01 | 0.44 ±0.02 | 0.73 ±0.00 |
| ***Average*** | 0.026 | 0.317 | **0.813** | 0.288 | 0.250 | 0.467 | 0.486 | 0.424 | 0.729 |

**Tested LLMs**: To comprehensively evaluate our algorithm's effectiveness, we tested a diverse range of LLMs with varying parameter sizes, including Qwen2.5-1.5B (Hui et al., 2024), OPT-2.7B (Zhang et al., 2022), Llama3.2-3B (Meta AI, 2024), Qwen2.5-3B, Llama2-7B (Touvron et al., 2023), Mixtral-7B (Jiang et al., 2024a), Qwen2.5-7B, Llama-3.1-8B (Dubey et al., 2024), and Llama2-13B (Touvron et al., 2023).We evaluated our `Water-Probe` algorithm on all LLMs, testing its performance under various watermarking schemes and in scenarios without watermarks.

**`Watermark-Probe` Settings**: For our `Watermark-Probe` algorithm, detailed prompts are provided in the appendix C. To calculate the z-score, we repeat each detection experiment 3 times to compute the standard deviation. We set $\mu = 0.1$ for our experiments.

## 4.2 MAIN RESULTS

In Table 1, we present the average similarity and standard deviation obtained using `Watermark-Probe-v1` and `Watermark-Probe-v2` algorithms for identifying various LLMs under different watermarking conditions and non-watermarked scenarios. For all LLMs, the sampling temperature was set to 1, with the number of samples set to $10^4$. As evident from Table 1, `Watermark-Probe-v1` demonstrates high effectiveness for N-gram based watermarking but is not applicable to fixed-key-list based watermarking. In contrast, the `Watermark-Probe-v2` algorithm proves effective in identifying all watermarking algorithms tested. Additionally, even watermarking algorithms claiming to be distortion-free, such as Aar (Aaronson & Kirchner, 2022), DiPMark (Wu et al., 2023b), and $\gamma$-reweighting (Hu et al., 2023), they can be effectively identified by both versions of `Watermark-Probe`. Furthermore, our algorithm maintains low similarity for non-watermarked LLMs, ensuring minimal false positive rates. Additionally, we calculated the average similarity for different watermarking algorithms in Table 1 to demonstrate their detection confidence. Among these, the Aar algorithm is the most easily detectable due to its pronounced perturbation for individual keys. Lastly, given the same number of samples, the `Watermark-Probe-v1` algorithm yields more significant identification results for N-gram based watermarking algorithms compared to the `Watermark-Probe-v2` algorithm.

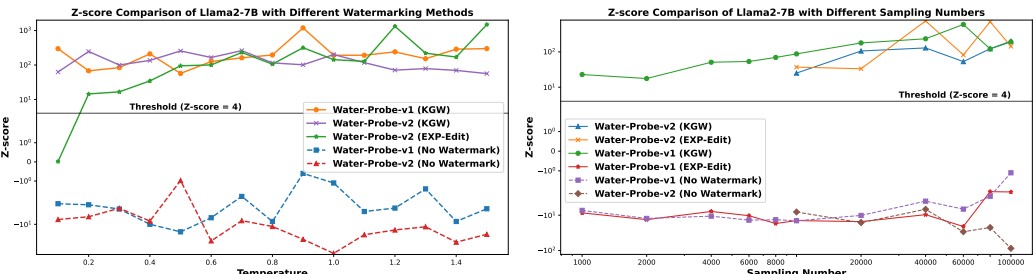

Figure 3: The left plot shows the variation of z-scores detected by `Watermark-Probe-v1` and `Watermark-Probe-v2` as a function of sampling temperature. The right plot illustrates the change in z-scores detected by `Watermark-Probe-v1` and `Watermark-Probe-v2` with different sampling numbers.

### 4.3 FURTHER ANALYSIS

**Influence of Sampling Temperature**: The results in Table 1 are based on a sampling temperature of 1. To further validate the performance of `Watermark-Probe` under different sampling temperatures, we show the z-score changes across temperatures in the left plot of Figure 3. Using Llama2-7B as an example, as the sampling temperature increases from 0.1 to 1.5, both `Watermark-Probe-v1` and `Watermark-Probe-v2` can distinguish between watermarked and unwatermarked LLMs. However, at relatively low temperatures ($T < 0.5$), detection of unwatermarked LLMs may show some fluctuations. Since deployed LLMs rarely use very low temperatures, our algorithm can be considered effective for detecting real-world LLM deployments.

**Influence of Sampling Number**: The right plot in Figure 3 illustrates the impact of the sampling number on the detected z-score. Specifically, we used Llama2-7B as the target LLM with a temperature of 1. We observed that `Watermark-Probe-v2` requires more samples compared to `Watermark-Probe-v1`. With insufficient samples, `Watermark-Probe-v2` lacks enough common prefixes to compute Equation 9. In our setting, `Watermark-Probe-v1` can achieve stable detection with 1,000 samples, while `Watermark-Probe-v2` requires at least $10^4$ samples. For cases where detection is successful, the z-score of watermarked LLMs tends to increase with the number of samples, although this trend exhibits fluctuations.

## 5 ENHANCING THE IMPERCEPTIBILITY OF WATERMARKED LLMs

We have demonstrated that current watermarked LLMs can be identified by our `Water-Probe` method. In this section, we discuss how to improve the imperceptibility of watermarked LLMs. The core principle is to make it challenging to construct repeated sampling scenarios using two separate keys according to Equation 9.

One design is **globally fixed watermark key** (e.g., Unigram (Zhao et al., 2023)). While it's easy to construct repeated sampling scenarios with a single key, we cannot detect stable deviations between different keys as only one exists globally. In the appendix E, we provide an algorithm named `Water-Contrast` to identify watermarks by comparing the target LLM distribution and a prior distribution. While not theoretically guaranteed (*it's challenging to determine if this bias is from watermarking or inherent to the LLM*), it shows practical effectiveness. Meanwhile, Unigram watermarks are susceptible to cracking (Jovanović et al., 2024).

The second design aims to increase the randomness of watermark key selection, making it less dependent on N-grams. This approach makes it difficult to construct repeated sampling scenarios using the same key. For **Fixed-Key-List Based watermarking**, a viable strategy is to increase the length of the key list. Since the initial key position is random, increasing the key list length enhances the randomness of key selection.

Additionally, for **N-gram Based watermarking algorithms**, we propose an enhanced strategy called `Water-Bag`, which combines multiple master watermark keys into a key-bag with a key inversion mechanism.

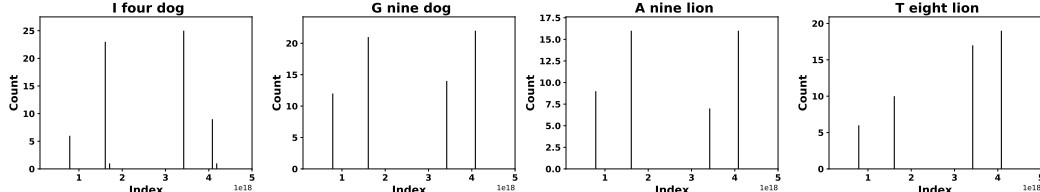

Figure 4: Distribution of start keys for identical prefixes in `Water-Bag` strategy. Analysis based on prompts described in Section 3.3 for `Water-Probe-v2`. Each subplot represents a specific prefix (showed in title).

Table 2: Performance comparison of `Water-Bag` Strategy and Exp-Edit algorithm in watermarked LLM identification and watermarked text detection. `Water-Bag` is evaluated with varying bag sizes, while Exp-Edit is tested with different key lengths. ▮ represents high-confidence watermark and ▮ represents low-confidence watermark. Detail of z-score could be seen in Appendix H.

| | None | KGW w. Water-Bag | | | | Exp-Edit(Key-len) | | |
| | | $|K \cup \overline{K}| = 1$ | $|K \cup \overline{K}| = 2$ | $|K \cup \overline{K}| = 4$ | $|K \cup \overline{K}| = 8$ | $|K| = 420$ | $|K| = 1024$ | $|K| = 2048$ |
|---|---|---|---|---|---|---|---|---|
| **Watermarked LLM Indentification** | | | | | | | | |
| Water-Probe-v1(n=3) | 0.02 ±0.01 | 0.42 ±0.01 | 0.05 ±0.01 | 0.02 ±0.01 | 0.03 ±0.02 | 0.03 ±0.05 | 0.02 ±0.01 | 0.02 ±0.02 |
| Water-Probe-v2(n=3) | 0.04 ±0.01 | 0.34 ±0.01 | 0.34 ±0.01 | 0.25 ±0.01 | 0.16 ±0.02 | 0.48 ±0.01 | 0.33±0.01 | 0.23±0.02 |
| Water-Probe-v2(n=5) | 0.06 ±0.06 | 0.32 ±0.01 | 0.18 ±0.01 | 0.12 ±0.01 | 0.07 ±0.01 | 0.64 ±0.00 | 0.54 ±0.01 | 0.44 ±0.00 |
| **Watermarked Text Detection** | | | | | | | | |
| Detection-F1-score | - | 1.0 | 1.0 | 1.0 | 1.0 | 1.0 | 0.975 | 1.0 |
| PPL | 8.15 | 11.93 | 11.85 | 12.17 | 12.50 | 16.63 | 17.28 | 19.06 |
| Robustness (GPT3.5) | - | 0.843 | 0.849 | 0.748 | 0.696 | 0.848 | 0.854 | 0.745 |
| Detection-time (s) | - | 0.045 | 0.078 | 0.156 | 0.31 | 37.87 | 108.5 | 194.21 |

**Definition 7** (Water-Bag Strategy). *The Water-Bag strategy extends N-gram based watermarking by using a set of master keys $K = \{K_1, K_2, ..., K_n\}$ and their inversions $\overline{K} = \{\overline{K_1}, \overline{K_2}, ..., \overline{K_n}\}$. For each generation, a master key $K_j$ or its inversion $\overline{K_j}$ is randomly selected:*

$$P_M^{WB}(y_i|x, y_{1:i-1}, K, \overline{K}) = F(P_M(y_i|x, y_{1:i-1}), k_i), \quad k_i = f(K_j^*, y_{i-n:i-1}), \quad K_j^* \sim Uniform(K \cup \overline{K}) \tag{11}$$

*where $P_M^{WB}$ is the modified probability distribution, $K_j^*$ is randomly sampled from the combined set of original and inverted keys, and $f$ is the watermark key derivation function. The inverted key $\overline{K_j}$ is defined as:*

$$\frac{1}{2}(F(P_M(y_i|x, y_{1:i-1}), f(K_j, y_{i-n:i-1})) + F(P_M(y_i|x, y_{1:i-1}), f(\overline{K_j}, y_{i-n:i-1}))) = P_M(y_i|x, y_{1:i-1}) \tag{12}$$

*This ensures that the average effect of $K_j$ and $\overline{K_j}$ on the logits is equivalent to the original logits, which makes our `Water-Probe-v1` ineffective against the Water-Bag strategy.*

For the watermarked text detection for `Water-Bag` Strategy, we use the maximum detection score across all master keys in the bag. The text is considered watermarked if this maximum exceeds a threshold.

To validate the effectiveness of the two strategies for enhancing the imperceptibility of watermarked LLMs, we evaluated their performance in Table 2. We examined both watermarked LLM identification and watermarked text detection settings, assessing the new watermarking strategies' detectability by our `Water-Probe` algorithm and their impact on watermarked text detection efficacy and performance.

For watermarked text detection, we used OPT-2.7B to generate texts on the C4 dataset (Raffel et al., 2020), using 30 tokens as prompts and generating 200 additional tokens with watermarks. PPL was calculated using Llama2-7B. To assess detection robustness, we computed the F1-score after rewriting texts using GPT-3.5. Detection time for single text was also recorded. We use the KGW algorithm as an implementation example for the `water-bag` strategy. For watermarked LLM identification, we report results for both `Water-Probe-v1` and `Water-Probe-v2`. For `Water-Probe-v2`, we present results for $n = 3$ and $n = 5$, where $n$ is the number of random tokens generated as described in Section 3.3. The $n = 3$ setting matches Table 1, while prompts for $n = 5$ are provided in Appendix C.

Table 2 demonstrates that `Water-Probe-v1` fails to effectively identify both `Water-Bag` and Exp-Edit algorithms. For `Water-Probe-v2`, detection difficulty increases with larger bag sizes in `Water-Bag`

and longer key lengths in Exp-Edit. However, `Water-Bag` proves more challenging to identify. Crucially, `Water-Bag`'s detectability remains stable as $n$ increases, while Exp-Edit becomes more easy to identify. We analyzed the key distribution of `Water-Bag` under `Water-Probe-v2` for different prefixes in Figure 4. The distribution is notably more uniform compared to Exp-Edit in Figure 2, explaining why `Water-Bag` is relatively harder to identify. We observe that increasing watermark key randomness reduces watermark robustness for both strategies, with Exp-Edit also significantly increasing detection time. This highlights a trade-off in watermarking algorithms between key randomness and robustness (Liu et al., 2024b). Future work should focus on developing algorithms that enhance randomness without compromising robustness.

## 6 RELATED WORK

Large language model (LLM) watermarking techniques (Liu et al., 2024b) have become crucial for copyright protection (Sander et al., 2024), generated text detection (Wu et al., 2023a), and preventing misuse (Liu et al., 2024c). LLM watermarking can be broadly categorized into two types. The first type is **post-processing watermarking**, which modifies generated text using format-based (Sato et al., 2023), lexical-based, syntax-based (Wei et al., 2022), or generation-based (Zhang et al., 2024) approaches to add watermark. However, post-processing watermarking methods require waiting for text generation to complete before modifying and adding watermarks, which is not suitable for current LLM services that require real-time text generation. Another category of watermarking algorithms, known as **watermarking during generation**, typically involves adjusting the distribution of the next generated token based on a watermark key. For instance, the KGW (Kirchenbauer et al., 2023a) algorithm divides the vocabulary into red and green lists, increasing the probability of tokens in the green list. SIR (Liu et al., 2024a) further modifies logits based on semantic information, enhancing watermark robustness. An important objective of these methods is to maintain the imperceptibility of generated text, i.e., watermarked and non-watermarked text should have identical distributions, with some distortion-free algorithms showing promising results (Kuditipudi et al., 2023; Aaronson & Kirchner, 2022). However, previous work has overlooked the imperceptibility of watermarked LLMs themselves, i.e., whether external users can detect if an LLM service contains watermarks without disclosure. This work investigates the imperceptibility of watermarked LLMs. The most relevant work is (Gloaguen et al., 2024), our work differs in several aspects: we propose a unified identification method for all **watermarking during generation** algorithms without requiring different detection approaches for different watermarks. Our method also supports identification of complex watermark variants (like EXP-Edit with sampling), and we introduce the `water-bag` algorithm to enhance watermarked LLM imperceptibility.

## 7 CONCLUSION

In this paper, we pioneered the study of identifying watermarked LLMs. We first theoretically demonstrated the basis for identifying watermarked LLMs. We then designed the `Water-Probe` algorithm, which identifies watermarked LLMs by comparing distribution differences of similar prompts under different watermark keys. Our experiments showed that our algorithm is applicable to all N-Gram and Fixed-Key-List based Watermarking algorithms, independent of sampling temperature. We discussed scenarios where `Water-Probe` might fail and designed the `WaterBag` watermarking algorithm, which sacrifices some robustness of watermarked text detection to make watermarked LLMs harder to identify. Future work could focus on watermark concealment as a key research direction, designing more covert watermarking schemes.

## 8 ACKNOWLEDGEMENTS

This work is primarily supported by the Key Research and Development Program of China (No. 2024YFB3309702). Additional support was provided by the National Science Foundation (NSF) under grants III-2106758 and POSE-2346158. This work was also supported by the Guangdong Provincial Department of Education Project (Grant No.2024KQNCX028); Scientific Research Projects for the Higher-educational Institutions (Grant No.2024312096), Education Bureau of Guangzhou Municipality; Guangzhou-HKUST(GZ) Joint Funding Program (Grant No.2025A03J3957), Education Bureau of Guangzhou Municipality.

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

# Part I

# Appendix

## Table of Contents

# A  PROOF OF THEOREM 1 AND ADDITIONAL ANALYSES

*Proof.* **Goal:** Prove that under the Lipschitz continuity condition, the watermark modification differences for two similar prompts $x_1$ and $x_2$ have high similarity, with an expected similarity of at least $\rho$.

**Step 1:** Express the watermark modification difference

For a prompt $x$ and keys $k_1, k_2$, define the watermark modification difference as:

$$\Delta_x(k_1, k_2) = P_M^F(\cdot|x, k_1) - P_M^F(\cdot|x, k_2) \tag{13}$$

**Step 2:** Utilize the Lipschitz continuity condition

By the Lipschitz continuity condition, for any fixed key $k$:

$$\|P_M^F(\cdot|x_1, k) - P_M^F(\cdot|x_2, k)\|_1 \leq L \cdot \|P_M(\cdot|x_1) - P_M(\cdot|x_2)\|_1 \tag{14}$$

Since $x_1$ and $x_2$ satisfy the similarity condition $\text{KL}(P_M(\cdot|x_1)||P_M(\cdot|x_2)) \leq \epsilon$ (Equation 5), by Pinsker's inequality (Csiszár & Körner, 2011):

$$\|P_M(\cdot|x_1) - P_M(\cdot|x_2)\|_1 \leq \sqrt{2\epsilon} \tag{15}$$

Therefore, for any $k \in \{k_1, k_2\}$:

$$\|P_M^F(\cdot|x_1, k) - P_M^F(\cdot|x_2, k)\|_1 \leq L \cdot \sqrt{2\epsilon} \tag{16}$$

**Step 3:** Analyze the difference of watermark modification differences

Consider the definition of $\Delta_x(k_1, k_2)$ and compute $\|\Delta_{x_1}(k_1, k_2) - \Delta_{x_2}(k_1, k_2)\|_1$:

$$
\begin{aligned}
\|\Delta_{x_1}(k_1, k_2) - \Delta_{x_2}(k_1, k_2)\|_1 &= \|(P_M^F(\cdot|x_1, k_1) - P_M^F(\cdot|x_1, k_2)) - (P_M^F(\cdot|x_2, k_1) - P_M^F(\cdot|x_2, k_2))\|_1 \\
&= \|(P_M^F(\cdot|x_1, k_1) - P_M^F(\cdot|x_2, k_1)) - (P_M^F(\cdot|x_1, k_2) - P_M^F(\cdot|x_2, k_2))\|_1 \\
&\leq \|P_M^F(\cdot|x_1, k_1) - P_M^F(\cdot|x_2, k_1)\|_1 + \|P_M^F(\cdot|x_1, k_2) - P_M^F(\cdot|x_2, k_2)\|_1 \\
&\leq L \cdot \sqrt{2\epsilon} + L \cdot \sqrt{2\epsilon} \\
&= 2L \cdot \sqrt{2\epsilon} \\
&= \delta'
\end{aligned}
\tag{17}
$$

where $\delta' = 2L \cdot \sqrt{2\epsilon}$.

**Step 4:** Relate $L_1$ distance to similarity measure

Assume $\text{Sim}(\cdot, \cdot)$ is cosine similarity, which is negatively correlated with $L_1$ distance. Since $\|\Delta_{x_1}(k_1, k_2) - \Delta_{x_2}(k_1, k_2)\|_1 \leq \delta'$, and $\delta'$ is a small positive number:

$$\text{Sim}(\Delta_{x_1}(k_1, k_2), \Delta_{x_2}(k_1, k_2)) \geq \rho' \tag{18}$$

where $\rho'$ is a lower bound dependent on $\delta'$, approaching 1 as $\delta'$ decreases.

**Step 5:** Calculate expected similarity

Since $\text{Sim}(\Delta_{x_1}(k_1, k_2), \Delta_{x_2}(k_1, k_2)) \geq \rho'$ for any $k_1, k_2 \in \mathcal{K}$, for randomly sampled $k_1, k_2$:

$$\mathbb{E}_{k_1, k_2 \sim \mathcal{K}}\left[\text{Sim}(\Delta_{x_1}(k_1, k_2), \Delta_{x_2}(k_1, k_2))\right] \geq \rho' \tag{19}$$

Set $\rho = \rho'$, and by choosing a sufficiently small $\epsilon$ (making $\delta'$ small enough), we can ensure $\rho$ is close to 1.

**Conclusion:** Under the Lipschitz continuity condition, for two similar prompts $x_1$ and $x_2$, the expected similarity of their watermark modification differences under randomly sampled watermark keys $k_1$ and $k_2$ is at least $\rho$, where $\rho$ is a large positive number close to 1. This proves the high consistency of watermark effects in similar contexts, ensuring the detectability and robustness of the watermark. $\square$

## B    STATISTICAL ANALYSIS OF UNWATERMARKED LLMS IDENTIFICATION

When there's no watermark present in the LLM, we expect the similarity measure in Equation 9 to be close to 0. This can be mathematically explained as follows:

For an unwatermarked model, $P_M^F = P_M$ for any $k$. Let $\hat{R}_N^{(1)}(\cdot)$ and $\hat{R}_N^{(2)}(\cdot)$ denote two independent empirical estimates of $R(\cdot)$ using $N$ samples each (because in the black-box setting, we cannot directly access the true $P_M^F$). Therefore:

$$
\begin{aligned}
\Delta_R(x_i, k_m, k_n) &= \hat{R}_N^{(1)}(P_M^F(\cdot|x_i, k_m)) - \hat{R}_N^{(2)}(P_M^F(\cdot|x_i, k_n)) \\
&= \hat{R}_N^{(1)}(P_M(\cdot|x_i)) - \hat{R}_N^{(2)}(P_M(\cdot|x_i)) \\
&= [\hat{R}_N^{(1)}(P_M(\cdot|x_i)) - R(P_M(\cdot|x_i))] - [\hat{R}_N^{(2)}(P_M(\cdot|x_i)) - R(P_M(\cdot|x_i))] \\
&= \epsilon_1 - \epsilon_2
\end{aligned}
\tag{20}
$$

where $\epsilon_1, \epsilon_2$ represent independent sampling errors that follow normal distributions $\mathcal{N}(0, \sigma^2)$ according to the Central Limit Theorem.

Consequently, for any $x_i, x_j, k_m, k_n$:

$$
\text{Sim}(\Delta_R(x_i, k_m, k_n), \Delta_R(x_j, k_m, k_n)) = \text{Sim}(\epsilon_1 - \epsilon_2, \epsilon_3 - \epsilon_4)
\tag{21}
$$

where $\epsilon_1, \epsilon_2$ represent independent sampling errors that follow normal distributions $\mathcal{N}(0, \sigma^2)$ according to the Central Limit Theorem.

Consequently, for any $x_i, x_j, k_m, k_n$:

$$
\text{Sim}(\Delta_R(x_i, k_m, k_n), \Delta_R(x_j, k_m, k_n)) = \text{Sim}(\epsilon_1 - \epsilon_2, \epsilon_3 - \epsilon_4)
\tag{22}
$$

where $\epsilon_1, \epsilon_2, \epsilon_3, \epsilon_4$ are independent sampling errors. To understand why this similarity has an expected value of zero, recall that cosine similarity is defined as:

$$
\text{Sim}(a, b) = \frac{a \cdot b}{||a|| \cdot ||b||}
\tag{23}
$$

Note that $(\epsilon_1 - \epsilon_2)$ and $(\epsilon_3 - \epsilon_4)$ are differences of independent normal variables, each following $\mathcal{N}(0, 2\sigma^2)$. For the numerator:

$$
\begin{aligned}
E[(\epsilon_1 - \epsilon_2)(\epsilon_3 - \epsilon_4)] &= E[\epsilon_1\epsilon_3] - E[\epsilon_1\epsilon_4] - E[\epsilon_2\epsilon_3] + E[\epsilon_2\epsilon_4] \\
&= E[\epsilon_1]E[\epsilon_3] - E[\epsilon_1]E[\epsilon_4] - E[\epsilon_2]E[\epsilon_3] + E[\epsilon_2]E[\epsilon_4] \\
&= 0
\end{aligned}
\tag{24}
$$

where the second equality follows from independence, and the final equality holds because $E[\epsilon_i] = 0$ for all $i$. As these differences are independent and orthogonal in expectation, their cosine similarity has an expected value of zero:

$$
\mathbb{E}[\text{Sim}(\epsilon_1 - \epsilon_2, \epsilon_3 - \epsilon_4)] = 0
\tag{25}
$$

Therefore, the average similarity $\bar{S}$ in Equation 9 has an expected value of 0.

## C    DETAILED PROMPTS FOR SIMULATING WATERMARK KEYS

In this section, we provide details of the prompt used for repeated sampling with the same key, as introduced in Section 3.3.

For the `Water-Probe-v1` algorithm, we use the following prompt pair as shown in Prompt 3 and Prompt 4.

---

**Prompt3: First** prompt for Fixed Key List Based Watermarking(`Water-Probe-v1`)

Please generate some text based on the following instructions(no other words):
First generate the prefix D seven tiger.
**Then answer the question: Name a country with a large population.**
Example1: D seven tiger China
Example2: D seven tiger India
Example3:

---

**Prompt4: First** prompt for Fixed Key List Based Watermarking(`Water-Probe-v1`)

Please generate some text based on the following instructions(no other words):
First generate the prefix D seven tiger.
**Then answer the question: Name a country with a large area.**
Example1: D seven tiger China
Example2: D seven tiger India
Example3:

---

Here, we use a fixed prefix D seven tiger as an example. In the actual experiment, we used 50 different prefixes to simulate 50 potentially different watermark keys. For the experiment in Table 1, we performed 10,000 samplings, with each prefix sampled 200 times. The complete list of 50 prefixes is shown below.

---

All the prefix for `Water-Probe-v1`

Y three lion, G three lion, U six lion, A eight tiger, K four cat, N seven tiger, K three cat, H five dog, E zero lion, V three dog, W five dog, K one tiger, B two tiger, E two lion, U six dog, A two tiger, D two tiger, I nine dog, F three lion, C three dog, N five cat, L two dog, K zero tiger, E five dog, B five cat, X four tiger, U three dog, K nine dog, P one dog, H zero dog, V eight tiger, S three tiger, P seven cat, S six dog, Y nine cat, J one tiger, C five tiger, A zero lion, L eight dog, X eight dog, I two dog, C eight tiger, O three tiger, L one cat, M five tiger, P five cat, F seven cat, I zero cat, P two lion, L four cat

---

**Prompt5: First** prompt for Fixed Key List Based Watermarking(`Water-Probe-v2`)

Please generate some text based on the following instructions(no other words):
The first word is randomly sampled from A-Z.
The second word is randomly sampled from zero to nine.
The third word is randomly sampled from cat, dog, tiger and lion.
Then add a separator | and **answer the following question: Name a country with a large population.**
Example1: D seven tiger | United States
Example2: J five dog | India
Example3: R six cat | China
Example4: T one tiger | Indonesia
Example5: P seven cat | Pakistan
Example6: G six cat | Russia
Example7: R five tiger | India
Example8: L nine cat | Mexico
Example9: T four tiger | United States
Example10: H three dog | Japan
Example11: B three lion | Germany
Example12:

---

---

**Prompt6: Second** prompt for Fixed Key List Based Watermarking(Water-Probe-v2)

Please generate some text based on the following instructions(no other words):
The first word is randomly sampled from A-Z.
The second word is randomly sampled from zero to nine.
The third word is randomly sampled from cat, dog, tiger and lion.
Then add a separator | and **answer the following question: Name a country with a large area.**
Example1: L three tiger | United States
Example2: X three cat | India
Example3: A six tiger | China
Example4: W eight lion | Argentina
Example5: D five dog | France
Example6: P one cat | Russia
Example7: E six tiger | Australia
Example8: Z eight lion | Canada
Example9: Q two tiger | United States
Example10: A nine cat | Brazil
Example11: V three dog | Russia
Example12:

---

Similarly, for Prompt 3 and Prompt 4, we assume that the answer spaces for the questions *Name a country with a large population.* and *Name a country with a large area* are highly correlated, i.e., countries with large areas tend to have relatively large populations. In practice, other correlated prompt pairs can be chosen, as long as they satisfy the correlation requirement.

For the Water-Probe-v2 algorithm, we use the following prompt pair shown in Prompt 5 and Prompt 6.

---

**Prompt7: First** prompt for Fixed Key List Based Watermarking(Water-Probe-v2 N=5)

Please generate some text based on the following instructions(no other words):
The first word is randomly sampled from A-Z.
The second word is randomly sampled from zero to nine.
The third word is randomly sampled from cat, dog, tiger and lion.
The fourth word is randomly sampled from apple, banana and orange.
The fifth word is randomly sampled from car, bus and truck.
The sixth entry is **the answer to the following question: Name a country with a large population.**
Example1: D seven tiger apple car United States
Example2: J five dog banana bus India
Example3: R six cat orange truck China
Example4: T one tiger apple bus Indonesia
Example5: P seven cat orange car Pakistan
Example6: G six cat banana truck Russia
Example7: R five tiger apple bus India
Example8: L nine cat banana car Mexico
Example9: T four tiger orange truck United States
Example10: H three dog apple bus Japan
Example11: B three lion orange truck Germany
Example12:

---

---

**Prompt8: Second** prompt for Fixed Key List Based Watermarking(Water-Probe-v2 N=5)

Please generate some text based on the following instructions(no other words):
The first word is randomly sampled from A-Z.
The second word is randomly sampled from zero to nine.
The third word is randomly sampled from cat, dog, tiger and lion.
The fourth word is randomly sampled from apple, banana and orange.
The fifth word is randomly sampled from car, bus and truck.
The sixth entry is **the answer to the following question: Name a country with a large population.**
Example1: D seven tiger apple car United States
Example2: J five dog banana bus India
Example3: R six cat orange truck China
Example4: T one tiger apple bus Argentina
Example5: P seven cat orange car France
Example6: G six cat banana truck Russia
Example7: R five tiger apple bus Australia
Example8: L nine cat banana car Canada
Example9: T four tiger orange truck United States
Example10: H three dog apple bus Brazil
Example11: B three lion orange truck Russia
Example12:

---

In this case, we selected the same question pair as in `Water-Probe-v1`. However, for `Water-Probe-v2`, all prefixes are randomly sampled by the LLM during the generation process. This random sampling by the LLM itself makes it easier to model the correlations between watermark keys at different positions.

Prompt 5 and Prompt 6 both generate prefixes of length 3. Since we further analyzed the case of generating prefixes of length 5 in Section 5, we also provide prompts for prefixes of length 5 in Prompt 7 and Prompt 8. It is worth noting that the longer the required prefix, the more total sampling times are needed, as more samples are required to cover all actually occurring prefixes. In this work, for the case where the prefix length is 3, we typically sampled 10,000 times, and for the case where the prefix length is 5, we typically sampled 100,000 times.

## D    DETAILED ALGORITHM FOR Water-Probe

To provide a clearer presentation of our Water-Probe algorithm, we present here a complete algorithmic representation, corresponding to the algorithm pipeline process described in Section 3.2. This algorithm flow can be used for both `Water-Probe-v1` and `Water-Probe-v2`, although the prompt construction process differs between them, as detailed in Appendix C.

## E    DETECTION OF GLOBAL-FIXED KEY WATERMARKING IN LLMS

In this section, we explore how to identify LLMs with global-fixed key based watermarking. As discussed in Section 5, since global-fixed key based watermarking uses only one key globally, we cannot compare differences between two different watermark keys. However, as the global-fixed key produces consistent bias across all prompts, we can calculate a prior distribution for each prompt and then verify if the differences from this prior distribution have high similarity across all prompt lists satisfying Equation 5.

Specifically, let $P_1, P_2, ..., P_N$ be the constructed prompt list, and $P_{\text{prior}}$ be their prior distribution. We can modify the average similarity calculation in Equation 9 as follows:

---

**Algorithm 1** Water-Probe Algorithm

---

**Require:** LLM $M$, significance level $\alpha$, sampling count $W$
**Ensure:** Watermark detection result
  1: Construct highly correlated prompts $P_1, P_2, ..., P_N$ satisfying Equation 5
  2: Define watermark key list $K = \{k_1, k_2, ..., k_m\}$
  3: **for** each prompt $P_i$ and key $k_j \in K$ **do**
  4:     Initialize count dictionary $C_{i,j}(y) \leftarrow 0$ for all $y \in V$
  5:     **for** $w = 1$ to $W$ **do**
  6:         Construct sampling prompt $P_{i,j}^w$ based on watermarking method (see Section 3.3)
  7:         Generate output $y_{i,j}^w \sim P_M^F(\cdot|P_{i,j}^w, k_j)$
  8:         $C_{i,j}(y_{i,j}^w) \leftarrow C_{i,j}(y_{i,j}^w) + 1$
  9:     **end for**
10:     $\hat{P}_M^F(y|P_i, k_j) \leftarrow C_{i,j}(y)/W$ for all $y \in V$
11: **end for**
12: Apply rank transformation to all $\hat{P}_M^F(\cdot|P_i, k_j)$
13: Calculate average similarity $\bar{S}$ using Equation 9
14: Compute z-score: $z = (\bar{S} - \mu)/\sigma$
15: **if** $z > z_\alpha$ **then**
16:     **return** LLM is likely watermarked
17: **else**
18:     **return** No evidence of watermarking
19: **end if**

---

Table 3: Identification of global fixed key watermarking (e.g., Unigram watermarking) in LLMs using a prior distribution computed as the average output distribution of all eight LLMs. ▬ indicates high-confidence watermark identification and ▬ indicates low-confidence watermark identification while no color indicates no watermark identified.

| LLM | Similarity | | Z-Score | |
|-----|-----------|-----------|---------|-----------|
| | Unigram | Unwatermark | Unigram | Unwatermark |
| GPT2 | 0.59 ±0.018 | 0.06 ±0.37 | 32.49 | 1.61 |
| OPT1.3B | 0.6 ±0.02 | 0.06 ±0.029 | 28.77 | 2.07 |
| OPT2.7B | 0.65 ±0.009 | 0.03 ±0.048 | 74.81 | 0.63 |
| LLama2-7B | 0.48 ±0.042 | 0.04 ±0.043 | 11.31 | 0.92 |
| LLama-2-13B | 0.38 ±0.037 | 0.08 ±0.018 | 10.17 | 4.54 |
| LLama-3.1-8B | 0.68 ±0.06 | 0.12 ±0.05 | 12.09 | 2.42 |
| Mixtral-7B | 0.87 ±0.056 | 0.038 ±0.052 | 22.24 | 0.74 |
| Qwen2.5-7B | 0.53 ±0.058 | 0.09 ±0.028 | 8.96 | 3.29 |

Table 4: Identification of global fixed key watermarking in LLMs using a single proxy model's output as prior distribution. ▮ indicates high-confidence watermark identification and ▮ indicates low-confidence watermark identification while no color indicates no watermark identified.

| LLM | GPT2 | | | | OPT1.3B | | | | OPT2.7B | | | | LLama2-7B | | | |
|---|---|---|---|---|---|---|---|---|---|---|---|---|---|---|---|---|
| | Similarity | | Z-Score | | Similarity | | Z-Score | | Similarity | | Z-Score | | Similarity | | Z-Score | |
| | W | NW | W | NW | W | NW | W | NW | W | NW | W | NW | W | NW | W | NW |
| GPT2 | $0.48_{\pm0.009}$ | $0.005_{\pm0.025}$ | 49.68 | 0.21 | $0.58_{\pm0.004}$ | $0.07_{\pm0.02}$ | 138.24 | 3.46 | $0.65_{\pm0.008}$ | $-0.07_{\pm0.015}$ | 77.60 | -0.49 | $0.42_{\pm0.017}$ | $0.034_{\pm0.015}$ | 25.37 | 2.25 |
| OPT1.3B | $0.71_{\pm0.07}$ | $0.09_{\pm0.04}$ | 9.433 | 2.31 | $0.59_{\pm0.007}$ | $0.04_{\pm0.02}$ | 81.44 | 1.84 | $0.67_{\pm0.007}$ | $0.05_{\pm0.015}$ | 92.46 | 3.43 | $0.45_{\pm0.008}$ | $0.11_{\pm0.04}$ | 53.21 | 2.55 |
| OPT2.7B | $0.7_{\pm0.03}$ | $0.1_{\pm0.05}$ | 23.89 | 1.70 | $0.59_{\pm0.01}$ | $0.07_{\pm0.03}$ | 53.05 | 2.41 | $0.64_{\pm0.007}$ | $0.025_{\pm0.025}$ | 88.79 | 0.99 | $0.46_{\pm0.04}$ | $0.03_{\pm0.01}$ | 11.16 | 2.47 |
| LLama2-7B | $0.66_{\pm0.015}$ | $0.1_{\pm0.007}$ | 44.03 | 14.08 | $0.57_{\pm0.025}$ | $0.07_{\pm0.02}$ | 22.38 | 3.32 | $0.67_{\pm0.008}$ | $0.028_{\pm0.023}$ | 79.72 | 1.21 | $0.43_{\pm0.02}$ | $0.06_{\pm0.02}$ | 18.35 | 2.60 |

---

**Prompt9: Used Prompt Set for Global-Fixed key Based Watermarking**

①: Please generate *random* *number* sequence between 0 to 9:2,3,9,8,0,4,7,5,6,1,5,8,7,1,2,4,6,0,9,3,

②: Please generate *random* *number* sequence between 0 to 9:3,8,0,7,4,5,6,1,9,2,1,3,7,9,2,0,4,6,8,5,

③: Please generate *random* *number* sequence between 0 to 9:5,8,7,1,2,4,6,0,9,3,2,3,9,8,0,4,7,5,6,1,

④: Please generate *random* *number* sequence between 0 to 9:0,7,2,3,6,5,1,9,8,4,3,8,0,7,4,5,6,1,9,2,

⑤: Please generate *random* *number* sequence between 0 to 9:1,3,7,9,2,0,4,6,8,5,0,7,2,3,6,5,1,9,8,4,

⑥: Please generate *random* *number* sequence between 0 to 9:6,0,9,3,2,3,9,8,0,4,7,5,6,1,5,8,7,1,2,4,

⑦: Please generate *random* *number* sequence between 0 to 9:4,6,0,9,3,2,3,9,8,0,4,7,5,6,1,5,8,7,1,2,

⑧: Please generate *random* *number* sequence between 0 to 9:9,8,0,4,7,5,6,1,5,8,7,1,2,4,6,0,9,3,1,3,

⑨: Please generate *random* *number* sequence between 0 to 9:7,4,5,6,1,9,2,1,3,7,9,2,0,4,6,8,5,0,7,2,

⑩: Please generate *random* *number* sequence between 0 to 9:8,0,7,4,5,6,1,9,2,5,8,7,1,2,4,6,0,9,3,2,

$$\bar{S} = \frac{1}{|\mathcal{P}|(|\mathcal{P}|-1)} \sum_{P_i \neq P_j \in \mathcal{P}} \text{Sim}(R(P_M(\cdot|P_i) - P_{\text{prior}}(\cdot|P_i)), R(P_M(\cdot|P_j) - P_{\text{prior}}(\cdot|P_j))). \quad (26)$$

For the prior distribution, we select N proxy LLMs and calculate the average output probability distribution under prompt $p_i$ as the prior distribution. Specifically:

$$P_{\text{prior}}(\cdot|p_i) = \frac{1}{N} \sum_{j=1}^{N} P_{M_j}(\cdot|p_i), \quad (27)$$

where $M_j$ represents the $j$-th proxy LLM, and $N$ is the total number of proxy LLMs used. Specifically, we used the following proxy LLMs: GPT2, OPT1.3B, OPT2.7B, Llama2-7B, Llama2-13B, Llama3-8.1B, Mixtral-7B, and Qwen2.5-7B.

Additionally, we utilized 10 distinct prompts for computation to validate our method as shown in Prompt 9.

The intuition behind this method is that if $M$ is watermarked, the differences between its output distributions and those of the proxy model should exhibit consistent patterns across different prompts, resulting in a high correlation. Conversely, for an unwatermarked model, we assume that the differences in bias between different language models are relatively small, and the correlation of these differences should be low.

Table 3 presents the identification results for global-fixed key watermarking using a prior distribution. Our method effectively identifies models employing global-fixed key watermarking, while yielding low z-scores for unwatermarked LLMs.

Table 5: Details of watermarking algorithms tested in our work.

| Algorithm Name | Category | Methodology |
|---|---|---|
| KGW (Kirchenbauer et al., 2023a) | N-Gram | Separate the vocabulary set into two lists: a red list and a green list based on the preceding token, then add bias to the logits of green tokens so that the watermarked text exhibits preference of using green tokens. |
| KGW-Min (Kirchenbauer et al., 2023b) | N-Gram | Similar to KGW, this approach partitions the vocabulary set based on the *minimum* token ID within a window of N-gram preceding tokens. |
| KGW-Skip (Kirchenbauer et al., 2023b) | N-Gram | Similar to KGW, this approach partitions the vocabulary set based on the *left-most* token ID within a window of N-gram preceding tokens. |
| Aar (Aaronson & Kirchner, 2022) | N-Gram | Generate a pseudo-random vector $r_t$ based on the N-gram preceding tokens to guide sampling at position $t$, and choose the token $i$ that maximize $r_t(i)^{1/p_t(i)}$ (exponential minimum sampling), where $p_t$ is the probability produced by LLM. |
| $\gamma$-Reweight (Hu et al., 2023) | N-Gram | Randomly shuffle the probability vector using a seed based on the preceding N-gram tokens. Discard the left half of the vector, doubling the remaining probabilities. Conduct further sampling using this reweighted distribution. |
| DiPmark (Wu et al., 2023b) | N-Gram | Similar to $\gamma$-Reweight, after shuffling, discard the left $\alpha$ portion of the vector and amplify the remaining probabilities by $1/(1-\alpha)$. |
| EXP-Edit (Kuditipudi et al., 2023) | Fixed-Key-List | Based on the Aar concept, construct a fixed pseudo-random vector list. When generating watermarked text, randomly select a start index in the list. For each watermarked token generation, sequentially use the pseudo-random vectors from this index for exponential minimum sampling. During detection, employ edit distance to calculate the correlation between the pseudo-random vector list and the text. |
| ITS-Edit (Kuditipudi et al., 2023) | Fixed-Key-List | Similar to EXP-Edit, this method also uses a fixed pseudo-random vector list. However, it uses inverse transform sampling instead of exponential minimum sampling during token selection. |

Table 6: Z-scores of waterbag method and Exp-Edit method. ▉ represents high-confidence watermark and ▉ represents low-confidence watermark, while no color means no watermark. This table provides supplementary information on the similarity content in Table 2.

| | | KGW w. Water-Bag | | | | Exp-Edit(Key-len) | | |
|---|---|---|---|---|---|---|---|---|
| | None | $|K \cup \overline{K}| = 1$ | $|K \cup \overline{K}| = 2$ | $|K \cup \overline{K}| = 4$ | $|K \cup \overline{K}| = 8$ | 420 | 1024 | 2048 |
| Watermarked LLM Indentification | | | | | | | | |
| Water-Probe-v1(n=3) | -8.20 | 11.67 | -4.22 | -7.84 | -4.34 | -3.07 | -5.87 | -5.17 |
| Water-Probe-v2(n=3) | -2.87 | 24.87 | 23.49 | 16.19 | 3.08 | 47.74 | 18.70 | 6.17 |
| Water-Probe-v2(n=5) | -100.43 | 28.56 | 13.68 | 1.38 | -4.41 | 131.30 | 78.63 | 69.40 |

To further validate the key factors in using prior distribution for testing, we conducted experiments using a single proxy model as the prior distribution, as shown in Table 4. We performed cross-experiments with different LLMs. These results demonstrate that global fixed-key watermarking can still be detected when using a single LLM as the prior distribution. However, the z-score detection for unwatermarked LLMs exhibits greater fluctuation. This is primarily due to the significant bias in a single proxy model as the prior distribution, affecting the variance of identification and resulting in small z-score.

Although our method using prior distribution achieved good results in our experiments, this identification approach has a limitation in that it can only detect stable biases in LLMs, assuming that a stable bias indicates a watermark. However, this assumption may not hold in real-world scenarios, as it is challenging to distinguish whether the bias is caused by watermarking or inherent to the LLM itself. This is particularly problematic in cases where LLMs are known to have inherent biases. Future work could investigate more interpretable detection methods.

## F    DETAILS OF TESTED WATERMARKING ALGORITHMS

To help understand the watermarking algorithms related to the experiments in this paper, we provide detailed information for all watermarking algorithms in Table 5, including their names, references, types, and brief descriptions. All our experiments were conducted using the MarkLLM (Pan et al., 2024) framework.

Table 7: Z-scores for watermark detection on various LLMs with and without watermarks using `Watermark-Probe-1` and `Watermark-Probe-2`. ▇ indicates high-confidence watermark identification, ▇ indicates low-confidence watermark identification, while no color indicates no watermark identified. This table provides supplementary information on the similarity content in Table 1.

| LLM | Non | N-Gram | | | | | | Fixed-Key-Set | |
| | | KGW | Aar | KGW-Min | KGW-Skip | DiPmark | $\gamma$-Reweight | EXP-Edit | ITS-Edit |
| --- | --- | --- | --- | --- | --- | --- | --- | --- | --- |
| **Watermark-Probe-1** (w. prompt 1) | | | | | | | | | |
| *Qwen2.5-1.5B* | -5.02 | 43.32 | 123.57 | 14.70 | 24.29 | 33.35 | 55.90 | -5.21 | -25.04 |
| *OPT-2.7B* | -5.99 | 49.21 | 117.95 | 16.69 | 35.78 | 93.25 | 49.98 | -1.32 | -1.27 |
| *Llama-3.2-3B* | -4.52 | 49.39 | 79.33 | 80.94 | 71.11 | 87.17 | 76.35 | -6.70 | -4.29 |
| *Qwen2.5-3B* | -6.70 | 48.20 | 127.07 | 14.73 | 625.36 | 56.15 | 42.96 | -6.18 | -1.97 |
| *Llama2-7B* | -8.20 | 30.01 | 109.00 | 25.51 | 29.75 | 44.35 | 106.37 | -3.07 | -28.05 |
| *Mixtral-7B* | -3.80 | 25.03 | 40.21 | 35.51 | 19.99 | 36.30 | 137.39 | -22.01 | -3.31 |
| *Qwen2.5-7B* | -1.16 | 38.25 | 30.03 | 22.85 | 50.34 | 47.31 | 50.48 | -3.39 | -15.92 |
| *Llama-3.1-8B* | -6.44 | 20.40 | 143.52 | 29.05 | 28.19 | 29.01 | 125.46 | -3.79 | -12.70 |
| *Llama2-13B* | -3.62 | 29.23 | 79.42 | 11.74 | 18.01 | 30.40 | 49.23 | -6.63 | -2.78 |
| **Watermark-Probe-2** (w. prompt 2) | | | | | | | | | |
| *Qwen2.5-1.5B* | -5.06 | 34.55 | 55.97 | 16.39 | 8.79 | 19.45 | 14.28 | 10.73 | 1840.44 |
| *OPT-2.7B* | -1.95 | 42.59 | 67.93 | 15.17 | 4.61 | 32.40 | 11.04 | 26.44 | 1073.13 |
| *Llama-3.2-3B* | -12.42 | 29.91 | 96.14 | 50.00 | 19.91 | 80.04 | 67.07 | 34.99 | 7702.12 |
| *Qwen2.5-3B* | -3.48 | 6.35 | 108.44 | 11.29 | 25.92 | 39.88 | 18.84 | 18.06 | 8209.12 |
| *Llama2-7B* | -2.87 | 24.87 | 40.05 | 32.68 | 15.62 | 35.50 | 25.11 | 47.74 | 6885.04 |
| *Mixtral-7B* | -0.87 | 6.09 | 54.39 | 11.14 | 13.49 | 49.02 | 111.12 | 14.83 | 1812.12 |
| *Qwen2.5-7B* | -2.48 | 8.90 | 185.88 | 10.50 | 13.06 | 7.64 | 12.40 | 13.04 | 1982.74 |
| *Llama-3.1-8B* | -64.31 | 25.24 | 104.77 | 10.03 | 49.23 | 38.47 | 81.36 | 31.35 | 12701.95 |
| *Llama2-13B* | -3.98 | 20.72 | 38.26 | 10.36 | 16.60 | 58.10 | 83.27 | 47.74 | 333.37 |

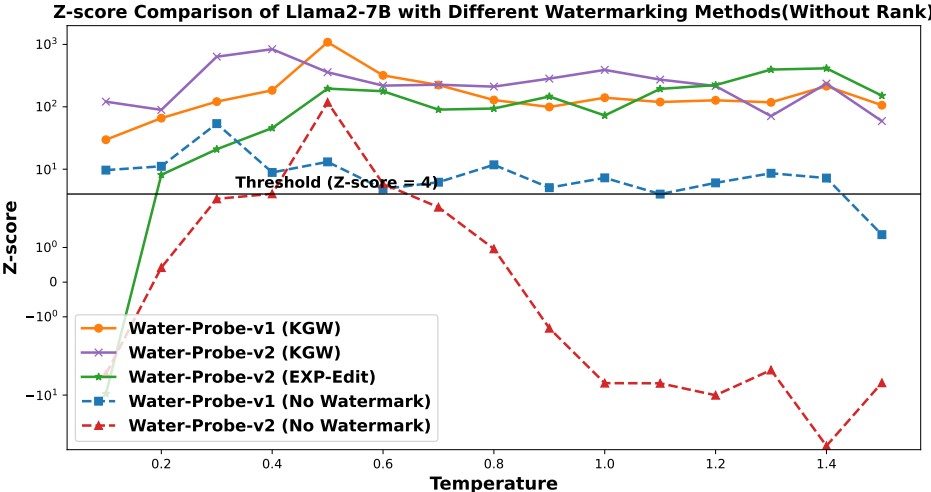

Figure 5: The variation of z-scores at different temperatures when calculating similarity without using rank transformation in Equation 9.

## G   ABLATION OF RANK TRANSFORMATION

To illustrate the importance of the rank transformation mentioned in Equation 9, we present in Figure 5 the variation of z-scores at different temperatures without using rank transformation. It can be observed that without rank transformation, the z-scores for Unwatermarked LLMs are significantly higher, especially at lower temperatures. Comparing the left plots in Figures 3 and 5, we can see that rank transformation effectively reduces the z-scores of Unwatermarked LLMs, making the identification and detection more stable.

Table 8: Identification p-value for various LLMs with different watermarks and without watermarks, using our two methods: `Watermark-Probe-v1` and `Watermark-Probe-v2`. ▇ represents high-confidence watermark and ▇ represents low-confidence watermark, while no color means no watermark. This table provides supplementary information on the similarity content in Table 1.

| LLM | | N-Gram | | | | | | Fixed-Key-Set | |
|-----|-----|-----|-----|-----|-----|-----|-----|-----|-----|
| | Non | KGW | Aar | KGW-Min | KGW-Skip | DiPmark | $\gamma$-reweight | EXP-Edit | ITS-Edit |
| **Watermark-Probe-v1 (w. prompt 1)** | | | | | | | | | |
| Qwen2.5-1.5B | 1 | 2.9e-410 | 5.9e-3319 | 3.2e-49 | 1.3e-130 | 3.6e-244 | 2.0e-681 | 1 | 1 |
| OPT-2.7B | 1 | 1.1e-528 | 3.4e-3024 | 7.7e-63 | 1.1e-280 | 2.6e-1891 | 2.9e-545 | 1-9.3e-2 | 1-1.0e-1 |
| Llama-3.2-3B | 1 | 1.6e-532 | 1.4e-1369 | 1.3e-1425 | 5.2e-1101 | 4.4e-1653 | 7.9e-1269 | 1 | 1 |
| Qwen2.5-3B | 1 | 2.7e-507 | 1.8e-3509 | 2.1e-49 | 8.3e-84925 | 1.7e-687 | 1.6e-403 | 1 | 1-2.4e-2 |
| Llama2-7B | 1 | 3.6e-198 | 4.3e-2583 | 7.6e-144 | 8.7e-195 | 7.0e-430 | 4.4e-2460 | 1-1.1e-3 | 1 |
| Mixtral-7B | 1 | 1.4e-138 | 8.0e-354 | 1.7e-276 | 3.4e-89 | 8.1e-289 | 3.9e-4102 | 1 | 1 |
| Qwen2.5-7B | 1-1.2e-1 | 2.1e-320 | 2.0e-198 | 7.3e-116 | 4.2e-553 | 7.9e-489 | 3.6e-556 | 1 | 1 |
| Llama-3.1-8B | 1 | 8.4e-93 | 4.4e-4476 | 7.7e-186 | 3.9e-175 | 2.5e-185 | 3.6e-3421 | 1 | 1 |
| Llama2-13B | 1 | 4.0e-188 | 1.1e-1372 | 4.0e-32 | 8.1e-73 | 2.7e-203 | 4.3e-529 | 1 | 1-2.7e-3 |
| **Watermark-Probe-v2 (w. prompt 2)** | | | | | | | | | |
| Qwen2.5-1.5B | 1 | 7.1e-262 | 4.1e-683 | 1.1e-60 | 7.5e-19 | 1.5e-84 | 1.5e-46 | 3.7e-27 | 9.8e-735530 |
| OPT-2.7B | 1-2.6e-2 | 1.2e-396 | 5.6e-1005 | 2.8e-52 | 2.0e-6 | 1.4e-230 | 1.2e-28 | 2.4e-154 | 1.2e-250072 |
| Llama-3.2-3B | 1 | 7.3e-197 | 3.5e-2010 | 1.1e-545 | 1.7e-88 | 3.7e-1394 | 9.2e-980 | 1.6e-268 | 2.5e-12881755 |
| Qwen2.5-3B | 1 | 1.1e-10 | 1.2e-2556 | 7.4e-30 | 2.0e-148 | 4.4e-348 | 1.8e-79 | 3.3e-73 | 7.3e-14633482 |
| Llama2-7B | 1-2.1e-3 | 7.9e-137 | 4.9e-351 | 1.5e-234 | 2.7e-55 | 2.5e-276 | 1.9e-139 | 1.0e-497 | 4.3e-10293604 |
| Mixtral-7B | 1-1.9e-1 | 5.6e-10 | 3.1e-645 | 4.0e-29 | 9.0e-42 | 1.3e-524 | 2.0e-2684 | 4.7e-50 | 6.5e-713068 |
| Qwen2.5-7B | 1-6.6e-3 | 2.8e-19 | 3.9e-7506 | 4.3e-26 | 2.8e-39 | 1.1e-14 | 1.3e-35 | 3.6e-39 | 3.1e-853666 |
| Llama-3.1-8B | 1 | 7.3e-141 | 1.0e-2386 | 5.6e-24 | 4.3e-529 | 4.5e-324 | 2.0e-1440 | 4.9e-216 | 7.5e-35034440 |
| Llama2-13B | 1 | 1.1e-95 | 1.4e-320 | 1.9e-25 | 3.5e-62 | 6.8e-736 | 1.0e-1508 | 1.0e-497 | 2.0e-24136 |

# H    SUPPLEMENTARY Z-SCORES AND P-VALUES

To facilitate a better understanding of the statistical methods used in identifying watermarked LLMs, we provide detailed information including z-scores and p-values for Table 1 and Table 2 in this section.

Specifically, Table 6 provides supplementary z-score information for Table 2, Table 7 provides supplementary z-score information for Table 1, and Table 8 provides supplementary p-value information for Table 1.

For all experiments, we consider a z-score below 4 to indicate no watermark, between 4 and 10 to indicate a watermark with relatively low confidence, and above 10 to indicate a watermark with high confidence.

# I    COMPARISON WITH RELATED WORK (GLOAGUEN ET AL., 2024)

1. **Universal Detection:** Our method (particularly Water-Probe-v2) can detect all current watermarking-during-generation approaches (those that modify generation logits or sampling processes). In contrast, Gloaguen et al. (2024)'s method requires specific designs for different watermarking algorithms:

   - Monte Carlo permutation test for red-green watermarking
   - Mann-Whitney U test for EXP-edit watermarking
   - Potential new methods for future watermarking methods

   Our approach represents the first universal detection method effective across all current LLM watermarking techniques.

2. **Unified Theoretical Foundation:** We provide a unified theoretical analysis and explanation for why watermarked LLMs can be detected, specifically demonstrating how watermark key conflicts lead to identifiable characteristics in model outputs. This theoretical framework provides a comprehensive understanding of the detection mechanism.

3. **Imperceptibility Enhancement:** Beyond detection methods, we also contribute the `Water-Bag` approach for improving the imperceptibility of watermarked LLMs, demonstrating significant improvements in watermark concealment while maintaining detectability.

Table 9: Experimental Results with Argmax Sampling On Exp-Edit

| Temp | Watermark F1 | Perplexity | Gloaguen P-value | Water-Probe-v2 P-value |
|------|------|------|------|------|
| 0.2 | 0.664 | 11.8 | <0.001 | <0.001 |
| 0.3 | 0.666 | 11.5 | <0.001 | <0.001 |
| 0.4 | 0.678 | 11.2 | <0.001 | <0.001 |
| 0.5 | 0.793 | 10.9 | <0.001 | <0.001 |
| 0.6 | 0.907 | 10.7 | <0.001 | <0.001 |
| 0.7 | 0.965 | 10.5 | <0.001 | <0.001 |
| 0.8 | 0.987 | 10.4 | <0.001 | <0.001 |
| 0.9 | 0.987 | 10.3 | <0.001 | <0.001 |
| 1.0 | 0.987 | 10.3 | <0.001 | <0.001 |

Table 10: Experimental Results with Multinomial Sampling On Exp-Edit, which is the most challenging case for watermarked LLM identification.

| Temp | Watermark F1 | Perplexity | Gloaguen P-value | Water-Probe-v2 P-value |
|------|------|------|------|------|
| 0.2 | 0.666 | 11.1 | <0.001 | <0.001 |
| 0.3 | 0.662 | 11.8 | 0.33 | <0.001 |
| 0.4 | 0.672 | 11.5 | 0.83 | <0.001 |
| 0.5 | 0.740 | 11.2 | 1.0 | <0.001 |
| 0.6 | 0.877 | 11.0 | 1.0 | <0.001 |
| 0.7 | 0.985 | 10.8 | 1.0 | <0.001 |
| 0.8 | 0.985 | 10.7 | 1.0 | <0.001 |
| 0.9 | 0.987 | 10.6 | 1.0 | <0.001 |
| 1.0 | 0.987 | 10.6 | 1.0 | <0.001 |

4. **Broader Applicability for Challenging Watermarking Variants:** Our method supports more challenging watermarking variants. For instance, while Gloaguen et al. (2024)'s experiments with EXP-edit only considered argmax sampling after exponential transformation (limiting a length-N watermark key list to at most N different sampling results), our method requires no such assumptions.

To demonstrate the broader applicability of our method, we conducted experiments with EXP-edit using sampling after exponential transformation. Given logits $l_i$, we first compute probabilities $p_i$ through temperature scaling:

$$p_i = \frac{\exp(l_i/\tau)}{\sum_j \exp(l_j/\tau)} \tag{28}$$

where $\tau$ is the temperature parameter. While Gloaguen et al. (2024)'s analysis focused on the deterministic argmax sampling variant:

$$i^* = \arg\max_i (\xi_i^{(j)})^{1/p_i} \tag{29}$$

This deterministic approach has a fundamental limitation - for a watermark key list of length N, it can only produce at most N distinct outputs. We instead evaluate multinomial sampling from the distribution:

$$P(i) \propto (\xi_i^{(j)})^{1/p_i} \tag{30}$$

Our experiments used the MarkLLM framework with EXP-Edit watermarking (key length = 420), using OPT-1.3B as the base model and LLaMA-7B for perplexity calculations. For Gloaguen et al. (2024)'s testing method, we generated 1,000 text samples of length 200 tokens each. For Water-Probe-v2 testing, we generated 10,000 text samples of length 5 tokens each.

Tables 9 and 10 show that when applying multinomial sampling after EXP transformation, the watermark detection F1 scores and perplexity values remain largely unaffected. However, while our method maintains its effectiveness, Gloaguen et al. (2024)'s method fails to detect the watermark. This demonstrates the broader applicability of our approach to practical watermarking deployments.

## J GUIDELINES FOR CONSTRUCTING WATERMARKED LLM IDENTIFICATION PROMPTS

To help better understand our method, we provide guidelines for constructing watermarked LLM identification prompts, divided into three parts: question space design, answer space design, and implementation and verification protocols.

### J.1 OVERALL PROMPT STRUCTURE

The prompts in our identification method consist of two essential components - a prefix component and a question component. The specific requirements for each component will be detailed in subsequent sections. Here is a basic illustration of the structure:

---
**Basic Two-Component Prompt Structure**

**Input Prompt:** Please start your answer with "WXYZ" (prefix component) and then answer the question: What is a major city in Asia? (question component)
**Response:** WXYZ Tokyo
**Explanation:** The generated prefix would help to fix the watermark key, while the actual answer would reflect the model's response distribution (achieved by repeated sampling).

---

### J.2 QUESTION COMPONENT DESIGN

As described in Section 3.2 Step 1, we should first construct highly correlated prompts with significantly overlapping but non-identical answer spaces. This enables easy assessment of how potential watermark keys affect the answer spaces of different prompts.

Here is a list of criteria for selecting questions:

1. **Answer Space Similarity:** Select questions with overlapping but non-identical answer spaces. For example:
   - "Name a country with a large population"
   - "Name a country with a large area"
   - "Name a country with a high GDP"
   - "Name a country with rich natural resources"

   These questions typically share common answers (e.g., USA, China, Russia) while maintaining distinct probability distributions over the answer space.

2. **Structural Requirements:**
   - Questions should be concise and unambiguous
   - Answers should come from a well-defined finite set (e.g., countries, cities)
   - Questions should maintain comparable difficulty levels
   - The target entity category should remain consistent within a test suite

### J.3 PREFIX COMPONENT DESIGN

### J.3.1 WATERMARK-PROBE-V1 CONSTRUCTION

For Watermark-Probe-v1, simply instruct the LLM to generate a fixed prefix before answering the question through explicit prompt instructions.

Here are the design principles for the prefix component:

1. Use meaningless character sequences (e.g., "abcd", "wxyz") that have no semantic meaning in any language
2. Avoid any sequences that could form acronyms, abbreviations or meaningful patterns
3. Ensure the prefix is completely unrelated to any potential answers or question domains
4. Keep the prefix length sufficient for determining the watermark key while maintaining semantic independence

Here is an example of the prefix component:

---

**Implementation Example for `Watermark-Probe-v1`**

Please generate *abcd* before answering the question.
**Question:** Name a country with a large population.
**Answer:** *abcd* India
**Explanation:** The generated prefix is meaningless and unrelated to the question domain, ensuring that it does not introduce any contextual bias.

---

### J.3.2 WATERMARK-PROBE-V2 CONSTRUCTION

For Watermark-Probe-v2, we need to design a controlled randomization process before answering the question to help fix the watermark key (see Section 3.3 for detailed reasons).

Here are the design principles for the prefix component:

1. Ensure the prefix generation does not influence the answer to the main question
2. Design multiple choice sets with logically equivalent probabilities
3. Keep the number of choices moderate and manageable
4. Maintain clear boundaries between different choice sets

Here is an example of the prefix component:

---

**Implementation Example for `Watermark-Probe-v2`**

Please generate a sentence that satisfies the following conditions:
- First word: Randomly sampled from A-Z
- Second word: Randomly sampled from zero to nine
- Third word: Randomly sampled from {cat, dog, tiger, lion}

Then answer: Name a country with a large population.
**Answer:** *A one cat* China
**Explanation:** All the possible generated prefixes are not related to the question domain, ensuring that they do not introduce any contextual bias.

---

## K THREAT MODEL

In this section, we outline the threat model under which our watermark identification method (detector) operates. We consider the capabilities and limitations of both the detector and the LLM service provider.

### K.1 DETECTOR CAPABILITIES

We assume the detector:

- Has black-box access to the LLM through standard API interfaces
- Can only interact with the model through normal prompt-response queries
- Has no access to model architecture, parameters, or training data
- Can perform multiple queries
- Cannot modify or influence the model's internal state

### K.2 Trust Assumptions

The threat model assumes:

- The LLM service provider may embed watermarks in the model outputs
- The API interface itself is trustworthy and returns genuine model outputs
- No man-in-the-middle attacks or response tampering occurs
- The detection process does not require knowledge of specific watermarking algorithms

### K.3 Detection Goals and Constraints

The primary objectives within this threat model are:

- Determine the presence or absence of watermarks in model outputs
- Maintain detection accuracy across different sampling temperatures and model configurations

Key constraints include:

- Detection must be performed solely through black-box testing
- Watermark removal or tampering is outside the scope
- Detection methods must be robust against normal model output variations

## L Test on Closed-source Models

We evaluated `Water-Probe-V2`'s detection capabilities on several closed-source models, including GPT-4o-mini, GPT-4o, GPT-3.5-turbo, Gemini-1.5-flash, and Gemini-1.5-pro. For all experiments, we utilized the latest API versions of these models (as of November 15, 2024) with a temperature setting of 0.7.

Table 11: Watermarked LLM Identification Results on Closed-source Models

| Model | Similarity | Std Dev | Z-score | Watermarked? |
|---|---|---|---|---|
| GPT-4o-mini | -0.005 | 0.018 | -5.984 | No |
| GPT-4o | 0.017 | 0.020 | -4.211 | No |
| GPT-3.5-turbo | 0.028 | 0.030 | -2.362 | No |
| Gemini-1.5-flash | 0.027 | 0.049 | -1.474 | No |
| Gemini-1.5-pro | 0.018 | 0.038 | -2.135 | No |

Our experimental results provide strong evidence that current closed-source model APIs do not contain watermarks. However, it is important to note that a key limitation of this experiment is our inability to verify ground truth labels, making it impossible to definitively confirm the accuracy of our detection results.

## M Reversion Key Calculation of Water-Bag

In this section, we provide detailed calculations for determining reversion keys that satisfy the constraints in Equation 11. Let $p = P_M(y_i|x, y_{1:i-1})$ represent the original model distribution, and $q = F(p, f(K_j, y_{i-n:i-1}))$ represent the distribution after modification using key $K_j$.

According to Equation 11, we have:

$$\frac{1}{2}(q + F(p, f(\overline{K_j}, y_{i-n:i-1}))) = p \tag{31}$$

Through algebraic manipulation, we can derive the required modification for the reversion key:

$$F(p, f(\overline{K_j}, y_{i-n:i-1})) = 2p - q \tag{32}$$

This equation provides the concrete method for calculating the reversion key $\overline{K_j}$. Specifically, for any input sequence $y_{i-n:i-1}$, the function $f(\overline{K_j}, y_{i-n:i-1})$ must map the original distribution $p$ to $2p - q$ to satisfy Equation 11.

It is important to note that a reversion key need not be restricted to numerical values. Any key that produces the required distributional modification qualifies as a valid reversion key, as long as it accurately satisfies the constraint equation.

