# OpenReview forum: "Can Watermarked LLMs be Identified by Users via Crafted Prompts?"
_ICLR.cc/2025/Conference — ICLR 2025 Spotlight_

### Official Review · Reviewer_AeYS · 2024-11-02

**Soundness:** 3
**Presentation:** 3
**Contribution:** 3
**Rating:** 8
**Confidence:** 3

**Summary:**

The paper proposes an identification algorithm to detect watermarks through well-designed prompts to LLMs, as current watermarked LLMs have consistent biases under the same watermark key.

**Strengths:**

The paper identifies watermarks in a black-box setting. Also, it proposes an effective method to improve watermark imperceptibility.

**Weaknesses:**

The primary concern I have with the paper is the similarity of distribution differences between prompts in the no-watermark case. First, cosine similarity measures the similarity between two non-zero vectors; thus, the equation Sim(0, 0) in the proof in Appendix B is confusing. Second, if we examine the distances of the elements directly rather than using the Sim function, the distances of the elements in the no-watermark case are approximately zero, suggesting high similarity.

The selection of prompts appears empirical. The paper could be strengthened by providing guidance on generating prompts that produce similar output distributions.

**Questions:**

Why is the similarity of distribution differences between prompts in the no-watermark case low?

---

### Official Review · Reviewer_qXQr · 2024-11-03

**Soundness:** 3
**Presentation:** 3
**Contribution:** 2
**Rating:** 6
**Confidence:** 4

**Summary:**

This paper addresses the imperceptibility of watermarking techniques used in Large Language Models (LLMs). It highlights that while existing watermarking techniques are effective in terms of detectability, minimal impact on text quality, and robustness to editing, they have not been thoroughly examined for their imperceptibility to users. The authors introduce an identification algorithm, Water-Probe, which is capable of detecting watermarks through carefully designed prompts. To enhance watermark imperceptibility, the authors propose a Water-Bag strategy, which merges multiple watermark keys to increase randomness.

**Strengths:**

- **Accuracy and Robustness**: The experimental results demonstrate the high accuracy and robustness of the Water-Probe algorithm across various LLMs, watermarking methods, and generation settings. The low false positive rate for non-watermarked LLMs further strengthens the algorithm’s reliability.
- **Practical Solution**: The proposed Water-Bag strategy offers a practical solution to improve the imperceptibility of watermarks, which is a critical concern for LLM providers.

**Weaknesses:**

- **Concept Confusion**: The manuscript mislead concept of watermarking and fingerprinting, the proposed method should be categorized into fingerprinting instead of watermarking. see Question 1.
- **Lack of Controbution**: The authors inputs prompts to see the response of watermarked LLM and non-watermarked LLM, which is so called identification algorithm. The manuscript just defeines some concept, samples prompt to see similarity of the inspected models.
- **Limited Scope of Water-Probe**: The Water-Probe algorithm might be limited in its application to only certain types of watermarking schemes. The paper does not sufficiently explore how the algorithm would perform against a wider range of watermarking techniques, especially those that are more sophisticated or less predictable.

**Questions:**

- **Concept Confusion**: the authors mislead watermarking concept.
LLM Watermarking is the process of injecting a invisible identifier into LLM, track unauthorized distribution, or indicate authenticity, withch contains two steps **injection** and **detection**. The author just talk about how to identify LLM output consistency, I never see any discussion about watermark injection in paper. The authors should clarify how their work relates to the injection and detection steps they have outlined, or explain why their approach focuses solely on identification without addressing injection.

- **Unknown Key Sampling Strategy**: the author propose the watermark identification method using setected prompts.
Step 1-3 only discuss the object of identification key sampling, never talk about the detail of identification key sampling.

- **Method Designed**: The author says “Research on the imperceptibility of watermarked LLMs should investigate whether watermarked and non-watermarked LLMs can be distinguished.”. However, in this paper, I never see the sampled prompts and identification method can solve this problem. The authors should clearly illustrate how the proposed method differentiates between watermarked and non-watermarked LLMs. I recommend including a specific experiment or analysis that directly addresses this assertion.

---

### Official Review · Reviewer_DykZ · 2024-11-03

**Soundness:** 3
**Presentation:** 3
**Contribution:** 3
**Rating:** 8
**Confidence:** 3

**Summary:**

The paper proposed a method called Water-Probe to detect whether a LLM is watermarked through well-designed prompts. Experiments show the method is effective to almost all mainstream watermarking algorithms. The authors also proposed a strategy to improves watermark imperceptibility by multiple keys.

**Strengths:**

1. It is reasonable to use repeated sampling to detect whether an LLM has been watermarked.
2. Prompts have been designed to reveal the connection between generated text and watermark keys in a black-box setting.
3. The evaluation is comprehensive, and many different watermarking methods have been tested. Experimental results also show that the proposed method works.

**Weaknesses:**

1. Symbol reuse. In Section 3 and Section 4, the symbol $P$ represents both the model distribution in $P_M$ and the prompt in $P_1, P_2, ..., P_N$, which can be confusing. Too many $P$ across line 222 to line 227.
2. Line 289, Figure 2. ‘KTH’ does not refer to anything else in other contexts. Is it Exp-Edit or ITS-Edit?

**Questions:**

1. In Table 1, why Water-Probe-V2 has low detection confidence for KGW series watermarks?
2. Could you provide some experiments on watermark detection on closed-source models such as ChatGPT using your tool? Since the proposed method is a black-box setting, this should be achievable.
3. In the method for enhancing the key pseudorandomness of the Water-Bag Strategy, your use of the "set of master keys and their inversions" approach lacks theoretical support. Does "inversions" refer to bit flips, or does it refer to certain keys that satisfy the constraints of equation (12)? If it’s the former, then the claim of enhanced pseudorandomness has no basis; if it’s the latter, the paper does not explain how to derive the inversion key space that satisfies equation (12), and it seems that satisfying (12) does not have a direct relationship with the pseudorandomness of $K^*_j$. The experiments in Table 2 test methods against the attacks proposed by the authors, without directly testing the pseudorandomness of the sequences in the key space using suites like NIST SP 800-22. Ultimately, this seems to only indicate that the specific method enhances defense capabilities against the proposed attacks, rather than demonstrating that the enhancement of pseudorandomness leads to improved defense capabilities, which is inconsistent with the authors' previous claims.

---

> ### Author Response · Authors · 2024-11-16
> **Thanks for your careful review （part 1/2）**
>
> Thank you for your valuable suggestions. We believe your concerns can be addressed through clarification and additional details in our responses below.
>
> ### About Symbol reuse.
>
> > In Section 3 and Section 4, the symbol *P* represents both the model distribution in *P_M* and the prompt in *P₁, P₂, ..., P_N*, which can be confusing. Too many *P* across line 222 to line 227.
>
> Thank you for your careful reading. We agree that the symbol reuse was confusing. In the new version, we use x to represent prompts throughout the paper, including section 4 and Appendix A and B, to address this concern.
>
> ### Line 289, Figure 2. ‘KTH’ does not refer to anything else in other contexts. Is it Exp-Edit or ITS-Edit?
>
> Sorry, this refers to the exp-edit method. We mistakenly used KTH which appears in some papers. We have corrected this in the new version.
>
>
> ### In Table 1, why Water-Probe-V2 has low detection confidence for KGW series watermarks?
>
> This issue mainly relates to the degree of distribution perturbation when KGW method is used with repeated sampling using the same watermark key. Compared to methods like EXP and EXP-Edit, KGW makes smaller modifications to the original distribution:
>
> 1. Methods like EXP and EXP-Edit use exponential sampling which significantly biases towards certain tokens, causing larger distribution perturbations.
> 2. In contrast, KGW only increases probability weights for green tokens moderately, resulting in milder perturbations.
>
> To quantitatively demonstrate this, we designed the following comparative experiment:
>
> Experimental Setup:
> - Initial logits: 2000-dimensional vector sampled from normal distribution in range (-100, 100)
> - Evaluation metric: JS divergence (measuring distribution difference before and after watermarking)
>
> KGW Experiment:
> - Parameters: gamma = 0.5, delta = 2.0
> - Repeated 100 times:
>   1. Add watermark to logits
>   2. Convert to probability distribution using softmax
>   3. Calculate JS divergence with original distribution
> - Record average JS divergence across 100 trials
>
> AAR Experiment:
> - Repeated 100 times:
>   1. Generate random 2000-dimensional key vector
>   2. Apply exponential transform and watermark to probability distribution
>   3. Calculate JS divergence after normalization
>   4. Average over 1000 repetitions per experiment
> - Record mean of 100 average JS divergences
>
> Results:
> - KGW average JS divergence: 0.373102
> - AAR average JS divergence: 5.067022
>
> These results clearly demonstrate that KGW's distribution perturbation is significantly smaller than AAR method, which explains why Water-Probe-V2 has lower detection confidence for KGW series watermarks.
>
> ### Could you provide some experiments on watermark detection on closed-source models such as ChatGPT using your tool? Since the proposed method is a black-box setting, this should be achievable.
>
> Please refer to the first point in the response to all reviewers.
>
> ### About the "inversions" in Water-Bag Strategy
>
> Here, our "inversions" refers to certain keys that satisfy the constraints of equation (12). We apologize for not directly providing the calculation method for these keys in the paper. However, this calculation can be derived directly from equation (12) by examining how the Inversion Key modifies the distribution through P_M(y_i | x, y_{1:i-1}) and F(P_M(y_i | x, y_{1:i-1}), f(K_j, y_{i-n:i-1})).
>
> Specifically, let $P_M(y_i | x, y_{1:i-1}) = p$ represent the original model distribution, and $F(p, f(K_j, y_{i-n:i-1})) = q$ represent the distribution after modification using key $K_j$.
>
> According to equation (12), we have:
> $\frac{1}{2}(q + F(p, f(\overline{K_j}, y_{i-n:i-1}))) = p$
>
> Through simple algebraic manipulation, we can derive:
> $F(p, f(\overline{K_j}, y_{i-n:i-1})) = 2p - q$
>
> This formula provides the concrete method for calculating the Inversion Key $\overline{K_j}$ - for any input sequence $y_{i-n:i-1}$, the function $f(\overline{K_j}, y_{i-n:i-1})$ needs to map the original distribution $p$ to $2p - q$ to satisfy equation (12). This is what we mean by keys that satisfy the constraints.
>
> It's worth noting that an inversion key doesn't necessarily have to be a number - as long as we can accurately calculate its modification to the distribution, it qualifies as a valid key.
>
> We will provide a detailed explanation of how to calculate this inversion key in the appendix M.

---

> ### Author Response · Authors · 2024-11-16
> **Thanks for your careful review （part 2/2）**
>
> ### About the pseudorandomness of the key space
>
> > Table 2 test methods against the attacks proposed by the authors, without directly testing the pseudorandomness of the sequences in the key space using suites like NIST SP 800-22. Ultimately, this seems to only indicate that the specific method enhances defense capabilities against the proposed attacks, rather than demonstrating that the enhancement of pseudorandomness leads to improved defense capabilities, which is inconsistent with the authors' previous claims.
>
> Thank you for your valuable suggestion. To rigorously evaluate the pseudorandomness of watermarking schemes, we designed the following comparative experiments.
>
> We constructed a simplified binary LLM model that outputs probabilities of 0 and 1 as 0.5 without watermarking. Based on this, we applied both KGW (window size=1) and water-bag strategy (Number of keys = 8) for watermark embedding, generating binary sequences of length 20,480,000. We conducted comprehensive statistical tests on these sequences using the NIST SP 800-22 test suite, with results as follows:
>
> KGW Generated Sequence Test Results (Sequence Length: 20480000)
>
> | Statistical Test | P-value | Pass Ratio | Result |
> |-----------------|---------|------------|--------|
> | Frequency (Monobit) | 0.437274 | 20/20 | Pass |
> | Frequency Test within Block | 0.000000 | 20/20 | Fail |
> | Runs | 0.000000 | 0/20 | Fail |
> | Longest Run of Ones | 0.000000 | 0/20 | Fail |
> | Binary Matrix Rank | 0.162606 | 20/20 | Pass |
> | Discrete Fourier Transform | 0.000000 | 0/20 | Fail |
> | Non-overlapping Template | 0.000000 | 0/20 | Fail |
>
> Water-Bag Generated Sequence Test Results (Sequence Length: 20480000)
>
> | Statistical Test | P-value | Pass Ratio | Result |
> |-----------------|---------|------------|--------|
> | Frequency (Monobit) | 0.911413 | 20/20 | Pass |
> | Frequency Test within Block | 0.437274 | 20/20 | Pass |
> | Runs | 0.834308 | 20/20 | Pass |
> | Longest Run of Ones | 0.637119 | 20/20 | Pass |
> | Binary Matrix Rank | 0.911413 | 20/20 | Pass |
> | Discrete Fourier Transform | 0.122325 | 20/20 | Pass |
> | Non-overlapping Template | 0.162606 | 20/20 | Pass |
>
> The test results demonstrate that sequences generated by the Water-Bag strategy exhibit good randomness characteristics across all statistical tests, while sequences generated by KGW only pass some basic tests. This further validates our theoretical analysis: the Water-Bag strategy indeed enhances the pseudorandomness of watermark sequences through the alternating use of multiple keys.

---

> ### Comment · Reviewer_DykZ · 2024-11-22
> **I will raise my score based on the clarification.**
>
> Thanks for the clarification, which addresses some of my key concerns. I have revised my score accordingly.
> - The results show that the gpt series models do not contain watermarks is surprising, but as you mentioned, we do not know the GROUND TRUTH.
> - The use of the NIST SP 800-22 test suite effectively addresses my question about directly evaluating the pseudorandomness of the key space. The results demonstrate the improved randomness characteristics of the Water-Bag strategy compared to the KGW strategy, supporting the claim that enhanced pseudo-randomness contributes to stronger defense capabilities. While the sequence length used in the experiments is relatively short compared to practical applications, it is sufficient to illustrate the key differences between the two methods.

---

> > ### Author Response · Authors · 2024-11-22
> >
> > Thank you for your positive feedback. We appreciate your time and consideration in reviewing the manuscript.

---

### Official Review · Reviewer_GLo8 · 2024-11-06

**Soundness:** 4
**Presentation:** 4
**Contribution:** 4
**Rating:** 8
**Confidence:** 4

**Summary:**

Considering that “even if individual watermarked texts are imperceptible, the distribution of numerous watermarked texts may reveal whether the LLM is watermarked, especially when repeatedly sampling with the same watermark key”, this paper proposed Water-Probe to detect if an LLM service contains watermarks through well-designed prompts then introduced Water-Bag strategy to improve watermark imperceptibility by merging multiple watermark keys.

**Strengths:**

1 This work is the first study on the imperceptibility of watermarked LLMs.
2 This paper is well organized and written, make it easy to follow.
3 The experiments are conducted across different LLMs and with different sampling methods and temperature settings. The conclusion and discussion based on the evaluation results are clear.

**Weaknesses:**

The threat model should be further described, especially in terms of the prior knowledge  assumptions of the detector.

**Questions:**

How about the blackbox detection on LLMs in real world? What is the difficulty?

---

> ### Author Response · Authors · 2024-11-16
> **Thanks for your careful review**
>
> We appreciate your valuable suggestions. We believe your concerns can be addressed through clarification and additional details in our responses below.
>
> ### About The threat model
>
> > The threat model should be further described, especially in terms of the prior knowledge assumptions of the detector.
>
> Thank you very much for your suggestion. We have supplemented our threat model below and also added a detailed description of the threat model in Appendix K (highlighted in red text).
>
> Detector Capabilities:
>
> 1. Can only access LLM service API as a normal user
> 2. Can only get outputs through normal prompt inputs
> 3. No access to model internal structure or parameters
> 4. No access to training data
> 5. Can make multiple queries
>
> Trust Assumptions:
>
> 1. LLM service provider may embed watermarks in outputs
> 2. API interface is trustworthy and returns actual model outputs
> 3. No man-in-the-middle attacks
>
> Attack Goals:
>
> 1. Detect presence of watermarks in outputs
> 2. Identify watermark type and characteristics where possible
>
> Constraints:
>
> 1. Black-box testing only
> 2. No watermark removal or tampering
> 3. No watermark injection during model training
>
> ### How about the blackbox detection on LLMs in real world? What is the difficulty?
>
> Please refer to the first point in the response to all reviewers.

---

### Public Comment · ~Nikola_Jovanović1 · 2024-11-15
**Missed Similar Prior Work**

Dear authors, it is good to see studies on underexplored aspects of LLM watermarking. However, I would like to point out that this submission overlooks prior work of Gloaguen et al. (NextGenAISafety @ ICML 2024, https://arxiv.org/abs/2405.20777). As it seems that you study the same problem and propose similar methods, I believe it is important to position your novelty claims in the context of this prior work. In particular, this undermines the central statements of the paper such as "_This work is the first to investigate the imperceptibility of watermarked LLMs, i.e., whether users can know if an LLM service is watermarked._".

---

> ### Author Response · Authors · 2024-11-15
> **Thank you for your reminder (Part 1/3)**
>
> Thank you for bringing up this related work. We apologize for missing Gloaguen et al.'s work in our initial literature review (our work began before June 2024). We became aware of their paper in early October when we noticed **it was also submitted to ICLR 2025**. If we had intended to deliberately omit citing their work, we would not have chosen to submit to ICLR, which has an open review process that encourages open comment discussions. After carefully reviewing their paper, we acknowledge their valuable contributions to this field and have thoroughly considered the distinctions between our works, which we will elaborate on below.
>
> In the following, we will first discuss the key distinctions between our works, and then provide experimental results to support our claims. Then there will be a discussion on ICLR's reviewer guidelines about concurrent work.
>
> Since our work was independently developed, our overall approach is quite different, and we believe our method has significant advantages, which we will discuss in detail below.
>
> ### Key distinctions between our works
>
> We believe it is important to acknowledge and discuss the relationship between our works. We have updated our introduction and related work sections in the revised version to properly contextualize our contributions alongside Gloaguen et al.'s work.
>
> Our work differs from Gloaguen et al.'s in four key aspects:
>
> 1. **Unified Detection Method vs. Specialized Detection Methods**
>    - Gloaguen et al. require different detection algorithms for different watermarking methods
>      - Monte Carlo permutation test for red-green watermarking
>      - Mann-Whitney U test for EXP-edit watermarking
>      - Potential new methods for future watermarking methods
>    - Our proposed Water-Probe-V2 is a unified detection method that:
>      * Effectively identifies all known watermarking-during-generation methods using a single detection approach
>      * Works on new watermarking methods without modification
>      * This universality stems from our deep understanding of watermarking mechanisms' fundamental characteristics
>
> 2. Our method provides a **unified theoretical analysis** and explanation for why watermarked LLMs can be detected, specifically showing how watermark key conflicts lead to identifiable patterns in watermarked LLMs.
>
> 3. Beyond just detecting watermarked LLMs, we also propose methods to **enhance their imperceptibility through our Water-Bag strategy**, achieving significant improvements.
>
> 4. Our method **supports more challenging watermarking method variants** . For instance, while Gloaguen et al.'s experiments with EXP-edit only used argmax sampling after exponential transformation (limiting an N-length watermark key list to at most N different sampling results), our method works **without such strong fixed sampling assumptions**. In Appendix K, we demonstrate experiments with EXP-edit using sampling after exponential transformation, comparing watermark detectability and LLM detectability, where only our method remains effective.

---

> ### Author Response · Authors · 2024-11-15
> **Thank you for your reminder (Part 2/3)**
>
> Specifically, regarding point 4, our work evaluates EXP-Edit [1] under a more general sampling scheme that better reflects real-world deployment scenarios. First, given logits $l_i$, we compute probabilities $p_i$ through temperature scaling:
>
> $p_i = \frac{\exp(l_i/\tau)}{\sum_j \exp(l_j/\tau)}$
>
> where $\tau$ is the temperature parameter. While Gloaguen et al.'s analysis focused on the deterministic argmax sampling variant of EXP-Edit:
>
> $i^* = \arg\max_i (\xi_i^{(j)})^{1/p_i}$
>
> This deterministic approach (given $\xi^{(j)}$) has a fundamental limitation - for a watermark key list of length N, it can only produce at most N distinct outputs. This severely restricts output diversity and makes watermarked LLM identification trivial, potentially compromising the watermark's security in practical deployments.
>
> We show here that simply doing multinomial sampling from the distribution:
>
> $P(i) \propto (\xi_i^{(j)})^{1/p_i}$
>
> would make the detection of watermarked LLMs significantly more difficult and do not sacrifice watermarked LLM's generation quality or detectability.
>
> Our experimental results demonstrate that while Gloaguen et al.'s detection method becomes ineffective under this most challenging sampling scheme, Water-Probe-V2 maintains robust detection capabilities, highlighting its broader applicability to practical watermarking deployments.
>
> Specifically, the following experiments used the MarkLLM [2] framework with EXP-Edit [1] watermarking (key length = 420), using OPT-1.3B as the base model and LLaMA-7B for PPL calculations. For Gloaguen's testing method, we generated 1000 text samples of length 200 tokens each. For Water-Probe-V2 testing, we generated 10000 text samples of length 5 tokens each (resulting in a smaller total token count).
>
> ### Argmax:
>
> | Temp | Watermark F1 | Perplexity | Gloaguen P-value | Water-Probe-V2 P-value |
> |------|--------------|------------|------------------|----------------------|
> | 0.2  | 0.664       | 11.8       | <0.001          | <0.001                |
> | 0.3  | 0.666       | 11.5       | <0.001          | <0.001                |
> | 0.4  | 0.678       | 11.2       | <0.001            | <0.001              |
> | 0.5  | 0.793       | 10.9       | <0.001            | <0.001              |
> | 0.6  | 0.907       | 10.7       | <0.001              | <0.001              |
> | 0.7  | 0.965       | 10.5       | <0.001            | <0.001              |
> | 0.8  | 0.987       | 10.4       | <0.001            | <0.001              |
> | 0.9  | 0.987       | 10.3       | <0.001             | <0.001              |
> | 1.0  | 0.987       | 10.3       |<0.001           | <0.001              |
>
> #### Multinomial:
>
> | Temp | Watermark F1 | Perplexity | Gloaguen P-value | Water-Probe-V2 P-value |
> |------|--------------|------------|------------------|----------------------|
> | 0.2  | 0.666       | 11.1       | <0.001            | <0.001                |
> | 0.3  | 0.662       | 11.8       |  0.33          | <0.001             |
> | 0.4  | 0.672       | 11.5       |  0.83           | <0.001              |
> | 0.5  | 0.740       | 11.2       |  1.0           |<0.001               |
> | 0.6  | 0.877       | 11.0       |  1.0           | <0.001              |
> | 0.7  | 0.985       | 10.8       |  1.0           | <0.001                |
> | 0.8  | 0.985       | 10.7       |  1.0           |<0.001                |
> | 0.9  | 0.987       | 10.6       |  1.0           |<0.001                |
> | 1.0  | 0.987       | 10.6       |  1.0           | <0.001              |
>
> From the experimental results above, we can observe that when applying Multinomial Sampling after EXP transformation, the watermark detection F1 scores and perplexity values remain largely unaffected. Our method remains effective, while Gloaguen et al.'s method fails to detect the watermark. This potentially indicates that our method has broader applicability.
>
> [1] Robust Distortion-free Watermarks for Language Models
>
> [2] MarkLLM: An Open-Source Toolkit for LLM Watermarking

---

> ### Author Response · Authors · 2024-11-15
> **Thank you for your reminder (Part 3/3)**
>
> ### This could be considered concurrent work under ICLR's guidelines
>
> Regarding the oversight, we would like to note ICLR's policy on recent work citations. According to ICLR's reviewer guidelines (https://iclr.cc/Conferences/2025/ReviewerGuide):
>
> > Q: Are authors expected to cite and compare with very recent work? What about non peer-reviewed (e.g., ArXiv) papers? (updated on 7 November 2022)
>
> > A: We consider papers contemporaneous if they are published within the last four months. That means, since our full paper deadline is October 1, if a paper was published (i.e., at a peer-reviewed venue) on or after July 1, 2024, authors are not required to compare their own work to that paper.
>
> The work in question was presented at workshop NextGenAISafety 2024 @ ICML on July 26, 2024 (https://icml-nextgenaisafety.github.io/). Under ICLR's guidelines, this would be considered contemporaneous work. Additionally, NextGenAISafety 2024 @ ICML has no proceedings, and **the paper is currently under submission to ICLR 2025**, making it a non-peer-reviewed work. By these criteria, it qualifies as concurrent work according to ICLR's policies.
>
> **We want to emphasize that our work was conducted completely independently, without any knowledge of or inspiration from Gloaguen et al.'s research. Our research began well before June 2024.**
>
>
> Finally, we deeply respect Gloaguen's work and are pleased to see other researchers working in this field. While we were not aware of Gloaguen's work at the time of our submission, we have included this discussion here. According to ICLR's guidelines, this can be considered concurrent work, which we acknowledge accordingly.
>
> **We have included this discussion in the revised version (introduction, related work and appendix I) and have changed our contribution claim based on Gloaguen's work. Thank you for your feedback.**

---

### Meta-Review · Area_Chair_R3na · 2024-12-18

**Metareview:**

This paper focuses on the imperceptibility of watermarking techniques in Large Language Models (LLMs). While existing methods ensure detectability, robustness, and minimal text quality degradation, their imperceptibility to users remains underexplored. The authors introduce Water-Probe, an identification algorithm that detects watermarks by using carefully designed prompts, proving effective against most mainstream watermarking methods. To address imperceptibility, the authors propose the Water-Bag strategy, which enhances randomness by merging multiple watermark keys. Experiments demonstrate the effectiveness of Water-Probe for watermark detection and the ability of Water-Bag to improve imperceptibility, advancing the evaluation and design of watermarking techniques for LLMs.

Overall, the problem is new and the approach is interesting. Since all the reviewers recommend acceptance; I also vote for acceptance.

**Additional Comments On Reviewer Discussion:**

The reviewer raised concerns about the closed model comparison and some minor issues. The authors clearly addressed these concerns, which led to an increase in the score. The average score is now 7.5, indicating a clear case for acceptance. I also vote for acceptance.

---

### Decision · Program_Chairs · 2025-01-22

Accept (Spotlight)